# ON THE WORD BOUNDARIES OF EMERGENT LANGUAGES BASED ON HARRIS'S ARTICULATION SCHEME

**Ryo Ueda, Taiga Ishii & Yusuke Miyao**
The University of Tokyo
{ryoryoueda,taigarana,yusuke}@is.s.u-tokyo.ac.jp

## ABSTRACT

This paper shows that emergent languages in signaling games lack meaningful word boundaries in terms of Harris's Articulation Scheme (HAS), a universal property of natural language. Emergent Languages are artificial communication protocols arising among agents. However, it is not obvious whether such a simulated language would have the same properties as natural language. In this paper, we test if they satisfy HAS. HAS states that word boundaries can be obtained solely from phonemes in natural language. We adopt HAS-based word segmentation and verify whether emergent languages have meaningful word segments. The experiment suggested they do not have, although they meet some preconditions for HAS. We discovered a gap between emergent and natural languages to be bridged, indicating that the standard signaling game satisfies prerequisites but is still missing some necessary ingredients.

## 1 INTRODUCTION

Communication protocols emerging among artificial agents in a simulated environment are called *emergent languages* (Lazaridou & Baroni, 2020). It is important to investigate their structure to recognize and bridge the gap between natural and emergent languages, as several structural gaps have been reported (Kottur et al., 2017; Chaabouni et al., 2019). For instance, Kottur et al. (2017) indicated that emergent languages are not necessarily compositional. Such gaps are undesirable because major motivations in this area are to develop interactive AI (Foerster et al., 2016; Mordatch & Abbeel, 2018; Lazaridou et al., 2020) and to simulate the evolution of human language (Kirby, 2001; Graesser et al., 2019; Dagan et al., 2021). Previous work examined whether emergent languages have the same properties as natural languages, such as grammar (van der Wal et al., 2020), entropy minimization (Kharitonov et al., 2020), compositionality (Kottur et al., 2017), and Zipf's law of abbreviation (ZLA) (Chaabouni et al., 2019).[1] *Word segmentation* would be another direction to understand the structure of emergent languages because natural languages not only have construction from word to sentence but also from phoneme to word (Martinet, 1960). However, previous studies have not gone so far as to address word segmentation, as they treat each symbol in emergent messages as if it were a "word" (Kottur et al., 2017; van der Wal et al., 2020), or ensure that a whole message constructs just one "word" (Chaabouni et al., 2019; Kharitonov et al., 2020).

The purpose of this paper is to study whether *Harris's articulation scheme* (HAS) (Harris, 1955; Tanaka-Ishii, 2021) also holds in emergent languages. HAS is a statistical universal in natural languages.[2] Its basic idea is that we can obtain word segments from the statistical information of phonemes but without referring to word meanings. HAS holds not only for phonemes but also for other symbol units like characters. HAS can be used for unsupervised word segmentation (Tanaka-Ishii, 2005) to allow us to study the structure of emergent languages. It should be appropriate to apply such unsupervised methods since word segments and meanings are not available beforehand in emergent languages.

---

[1] ZLA states that the more frequently a word is used, the shorter it tends to be (Zipf, 1935).
[2] Note that this is different from the famous *distributional* hypothesis (Harris, 1954).

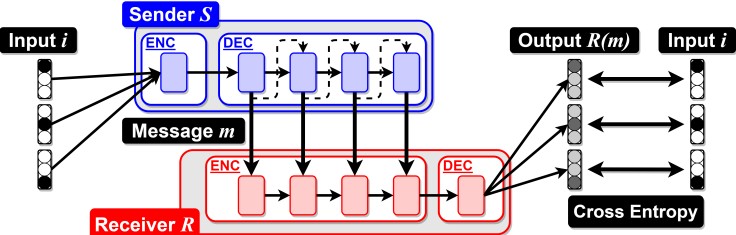

Figure 1: Illustration of a signaling game. Section 3.1 gives its formal definition. In each play, a sender agent obtains an input and converts it to a sequential message. A receiver agent receives the message and converts it to an output. Each agent is represented as an encoder-decoder model.

In addition to the absence of ground-truth data on segmentation, it is not even evident, in the first place, whether emergent languages have meaningful word segments. In other words, the problem is whether they have meaningful segments. If not, then it means that we find another gap between emergent and natural languages. In this paper, we pose several verifiable questions to answer whether their segments are meaningful. Importantly, some of the questions are applicable to any word segmentation scheme as well as HAS, so that they can be general criteria for future work on the segmentation of emergent languages.

To simulate the emergence of language, we adopt Lewis's signaling game (Lewis, 1969). This game involves two agents called *sender S* and *receiver R*, and allows only one-way communication from $S$ to $R$. In each play, $S$ obtains an input $i \in \mathcal{I}$ and converts $i$ into a sequential message $m = S(i) \in \mathcal{M}$. Then, $R$ receives $m \in \mathcal{M}$ and predicts the original input. The goal of the game is the correct prediction $S(m) = i$. Figure 1 illustrates the signaling game. Here, we consider the set $\{m \in \mathcal{M} \mid m = S(i)\}_{i \in \mathcal{I}}$ as the dataset of an emergent language, to which the HAS-based *boundary detection* (Tanaka-Ishii, 2005) is applicable. The algorithm yields the *segments* of messages.

Our experimental results showed that emergent languages arising from signaling games satisfy two preconditions for HAS: (i) the conditional entropy (Eq. 2) decreases monotonically, and (ii) the branching entropy (Eq. 1) repeatedly falls and rises. However, it was also suggested that the HAS-based boundaries are not necessarily meaningful. Segments divided by the boundaries may not serve as meaning units, while words in natural languages do (Martinet, 1960). It is left for future work to bridge the gap between emergent and natural languages in terms of HAS, by giving rise to meaningful word boundaries.

## 2   HARRIS'S ARTICULATION SCHEME

This section introduces Harris's articulation scheme (HAS) and the HAS-based boundary detection algorithm. HAS has advantages such as being simple, deterministic, and easy-to-interpret while being linguistically motivated. Such simplicity is important particularly when we have neither any prior knowledge nor ground-truth data of target languages, e.g., emergent languages.

In the paper "From phoneme to morpheme" (Harris, 1955), Harris hypothesized that word boundaries tend to occur at points where the number of possible successive phonemes reaches a local peak in a context. Harris (1955) exemplifies the utterance "He's clever" that has the phoneme sequence /hiyzclevər/.[3] The number of possible successors after the first phoneme /h/ is 9: /w,y,i,e,æ,a,ə,o,u/. Next, the number of possible successors after /hi/ increases to 14. Likewise, the number of possible phonemes increases to 29 after /hiy/, stays at 29 after /hiyz/, decreases to 11 after /hiyzk/, decreases to 7 after /hiyzkl/, and so on. Peak numbers are found at /y/, /z/, and /r/, which divides the phoneme sequence into /hiy/+/z/+/klevər/. Thus, the utterance is divided into "He", "s", and "clever". Harris's hypothesis can be reformulated from an information-theoretic point of view by replacing the *number* of successors with *entropy*. We review the mathematical formulation of the hypothesis as *Harris's articulation scheme* (HAS) and the HAS-based boundary detection (Tanaka-Ishii, 2005). HAS does involve statistical information of phonemes but does not involve word meanings. This is important because it gives a natural explanation for a well-known linguistic concept called *double articulation* (Martinet, 1960). Martinet (1960) pointed out that languages have two structures: phonemes (irrelevant to meanings) and meaning units (i.e., words and morphemes). HAS can construct meaning units without referring to meanings.

---

[3]There may be other representations for the phonemes, but we follow Harris's notation.

## 2.1 Mathematical Formulation of Harris's Hypothesis

While Harris (1955) focuses on phonemes for word boundary detection, Tanaka-Ishii (2021) suggests that the hypothesis is also applicable to units other than phonemes. Therefore, in this section, a set of units is called an *alphabet* $\mathcal{X}$ as a purely mathematical notion that is not restricted to phonemes. Tanaka-Ishii (2005) uses characters for the same purpose. Moreover, Frantzi & Ananiadou (1996) and Tanaka-Ishii & Ishii (2007) investigate the detection of collocation from words.

Let $\mathcal{X}$ be an alphabet and $\mathcal{X}^n$ be the set of all $n$-grams on $\mathcal{X}$. We denote by $X_i$ a random variable of $\mathcal{X}$ indexed by $i$, and by $X_{i:j}$ a random variable sequence from $X_i$ to $X_j$. The formulation by Tanaka-Ishii (2005) involves two kinds of entropy: *branching entropy* and *conditional entropy* (Cover & Thomas, 2006).[4] The *branching entropy of a random variable $X_n$ after a sequence $s = x_0 \cdots x_{n-1} \in \mathcal{X}^n$* is defined as:

$$h(s) \equiv \mathcal{H}(X_n \mid X_{0:n-1} = s) = -\sum_{x \in \mathcal{X}} P(x \mid s) \log_2 P(x \mid s), \qquad (1)$$

where $P(x \mid s) = P(X_n = x \mid X_{0:n-1} = s)$. Intuitively, the branching entropy $h(s)$ means how many elements can occur after $s$ or the uncertainty of the next element after $s$. In addition to $h(s)$, the *conditional entropy of a random variable $X_n$ after an $n$-gram sequence $X_{0:n-1}$* is defined as:

$$H(n) \equiv \mathcal{H}(X_n \mid X_{0:n-1}) = -\sum_{s \in \mathcal{X}^n} P(s) \sum_{x \in \mathcal{X}} P(x \mid s) \log_2 P(x \mid s), \qquad (2)$$

where $P(s) = P(X_{0:n-1} = s)$. The conditional entropy $H(n)$ can be regarded as the mean of $h(s)$ over $n$-gram sequences $s \in \mathcal{X}^n$, since $H(n) = \sum_{s \in \mathcal{X}^n} P(s)h(s)$. $H(n)$ is known to decrease monotonically in natural languages (Bell et al., 1990). Thus, for a partial sequence $x_{0:n-1} \in \mathcal{X}^n$, $h(x_{0:n-2}) > h(x_{0:n-1})$ holds on average, although $h(s)$ repeatedly falls and rises depending on a specific $s$. Based on such properties, *Harris's articulation scheme* (HAS) is formulated as:[5]

$$\text{If there is some partial sequence } x_{0:n-1} \in \mathcal{X}^n \ (n > 1)$$
$$\text{s.t. } h(x_{0:n-2}) < h(x_{0:n-1}), \text{ then } x_n \text{ is at a } boundary. \qquad (3)$$

## 2.2 Boundary Detection Algorithm Based on Harris's Articulation Scheme

This section introduces the HAS-based *boundary detection algorithm* (Tanaka-Ishii, 2005). Let $s = x_0 \cdots x_{n-1} \in \mathcal{X}^n$. We denote by $s_{i:j}$ its partial sequence $x_i \cdots x_j$. Given $s$ and a parameter *threshold*, the boundary detection algorithm yields boundaries $\mathcal{B}$.[6] It proceeds as shown in Algorithm 1. Entropy-based methods are advantageous in simplicity and interpretability, though they are inferior to other state-of-the-art methods in terms of F score.[7] Simplicity and interpretability are important when we analyze emergent languages for which we have no prior knowledge and have no ground-truth data of word segmentation. Note also that it is not even evident, in the first place, whether they have meaningful word segments.

---
**Algorithm 1** Boundary Detection Algorithm
---
1: $i \leftarrow 0$; $w \leftarrow 1$; $\mathcal{B} \leftarrow \{\}$
2: **while** $i < n$ **do**
3:     Compute $h(s_{i:i+w-1})$
4:     **if** $w > 1$ & $h(s_{i:i+w-1}) - h(s_{i:i+w-2}) >$ *threshold* **then**
5:         $\mathcal{B} \leftarrow \mathcal{B} \cup \{i + w\}$
6:     **end if**
7:     **if** $i + w < n - 1$ **then**
8:         $w \leftarrow w + 1$
9:     **else**
10:         $i \leftarrow i + 1$; $w \leftarrow 1$
11:     **end if**
12: **end while**
---

Since our targets are emergent languages, the outputs of the boundary detection algorithm do not necessarily mean articulatory boundaries. Instead, we call them *hypothetical boundaries (hypo-boundaries)* and refer to the segments split by hypo-boundaries as *hypo-segments*.

---

[4]The term "branching entropy" is from Tanaka-Ishii & Jin (2008), but the definition per se is quite basic.

[5]Although this is called *hypothesis* in Tanaka-Ishii (2005), Tanaka-Ishii & Jin (2006) and Tanaka-Ishii & Ishii (2007), we refer to it as *scheme* following the recent publication (Tanaka-Ishii, 2021).

[6]For practical reasons, the original algorithm involves another parameter *maxlen* to ensure $w < $ *maxlen*. We omit it because the message length in emergent languages is fixed in this paper (see Section 3).

[7]Tanaka-Ishii & Jin (2008) reported F-score $= 83.6\%$ in English and F-score $= 83.8\%$ in Chinese. They are considerably high scores for entropy-based settings.

## 3 EMERGENT LANGUAGE ARISING FROM SIGNALING GAME

We have to define environments, agent architectures, and optimization methods for language emergence simulations. This paper adopts the framework of Chaabouni et al. (2020). We define an environment in Section 3.1, specify the agent architecture and optimization methods in Section 3.2, and also give an explanation of the compositionality of emergent languages in Section 3.3.

### 3.1 SIGNALING GAME

An environment is formulated based on Lewis's signaling game (Lewis, 1969). *A signaling game $G$ consists of a quadruple $(\mathcal{I}, \mathcal{M}, S, R)$, where $\mathcal{I}$ is an *input space*, $\mathcal{M}$ is a *message space*, $S : \mathcal{I} \to \mathcal{M}$ is a *sender agent*, and $R : \mathcal{M} \to \mathcal{I}$ is a *receiver agent*. The goal is the correct reconstruction $i = R(S(i))$ for all $i \in \mathcal{I}$. While the input space $\mathcal{I}$ and the message space $\mathcal{M}$ are fixed, the agents $S, R$ are trained for the goal. An illustration of a signaling game is shown in Figure 1. Following Chaabouni et al. (2020), we define $\mathcal{I}$ as an attribute-value set $\mathcal{D}_{n_{\text{val}}}^{n_{\text{att}}}$ (defined below) and $\mathcal{M}$ as a set of discrete sequences of fixed length $k$ over a *finite alphabet* $\mathcal{A}$:

$$\mathcal{I} \equiv \mathcal{D}_{n_{\text{val}}}^{n_{\text{att}}}, \quad \mathcal{M} \equiv \mathcal{A}^k = \{a_1 \cdots a_k \mid a_j \in \mathcal{A}\}. \tag{4}$$

**Attribute-Value Set** Let $n_{\text{att}}, n_{\text{val}}$ be positive integers called *the number of attributes* and *the number of values*. Then, an *attribute-value set* $\mathcal{D}_{n_{\text{val}}}^{n_{\text{att}}}$ is the set of ordered tuples defined as follows:

$$\mathcal{D}_{n_{\text{val}}}^{n_{\text{att}}} \equiv \{(v_1, \ldots, v_{n_{\text{att}}}) \mid v_j \in \{1, \ldots, n_{\text{val}}\}\}. \tag{5}$$

This is an abstraction of an attribute-value object paradigm (e.g., Kottur et al., 2017) by Chaabouni et al. (2020). Intuitively, each index $j$ of a vector $(v_1, \ldots v_j, \ldots, v_{n_{\text{att}}})$ is an attribute (e.g., *color*), while each $v_j$ is an attribute value (e.g., *blue*, *green*, *red*, and *purple*).[8]

### 3.2 ARCHITECTURE AND OPTIMIZATION

We follow Chaabouni et al. (2020) as well for the architecture and optimization method. Each agent is represented as an encoder-decoder model (Figure 1): the sender decoder and the receiver encoder are based on single-layer GRUs (Cho et al., 2014), while the sender encoder and the receiver decoder are linear functions. Each element $i \in \mathcal{D}_{n_{\text{val}}}^{n_{\text{att}}}$ is vectorized to be fed into or output from the linear functions. Formally, each $i = (v_1, \ldots, v_{n_{\text{att}}})$ is converted into the $n_{\text{att}} \times n_{\text{val}}$-dimensional vector which is the concatenation of $n_{\text{att}}$ one-hot representations of $v_j$. During training, the sender samples messages probabilistically. At the test time, it samples them greedily to serve as a deterministic function. Similarly, the receiver's output layer, followed by the Softmax, determines $n_{\text{att}}$ categorical distributions over values $\{1, \ldots, n_{\text{val}}\}$ during training. At the test time, $n_{\text{att}}$ values are greedily sampled. The agents are optimized with the *stochastic computation graph* (Schulman et al., 2015) that is a combination of REINFORCE (Williams, 1992) and standard backpropagation. The sender is optimized with the former, while the receiver is optimized with the latter.

### 3.3 COMPOSITIONALITY OF EMERGENT LANGUAGES

An attribute-value set $\mathcal{D}_{n_{\text{val}}}^{n_{\text{att}}}$ by Chaabouni et al. (2020) is an extension of an attribute-value setting (Kottur et al., 2017) introduced to measure the compositionality of emergent languages. While the concept of compositionality varies from domain to domain, researchers in this area typically regard it as the *disentanglement* of representation learning. Kottur et al. (2017), for instance, set up an environment where objects have two attributes: *color* and *shape*, each of which has several possible values (e.g., *blue*, *red*, ... for color and *circle*, *star*, ... for shape). They assumed that if a language is sufficiently compositional, each message would be a composition of symbols denoting the color value and shape value separately. This concept has been the basis for subsequent studies (Li & Bowling, 2019; Andreas, 2019; Ren et al., 2020; Chaabouni et al., 2020).

---

[8]Although the game is extremely simple, it is suitable for avoiding some pitfalls. Lowe et al. (2019) indicated that agents might not communicate effectively in more complex games than in a signaling game. Bouchacourt & Baroni (2018) suggested that agents fail to capture conceptual properties when $\mathcal{I}$ is a set of images.

**Topographic Similarity**    *Topographic Similarity* (TopSim) (Brighton & Kirby, 2006; Lazaridou et al., 2018) is the de facto compositionality measure in emergent communication literature. Suppose we have distance functions $d_\mathcal{I}, d_\mathcal{M}$ for spaces $\mathcal{I}, \mathcal{M}$, respectively. TopSim is defined as the Spearman correlation between distances $d_\mathcal{I}(i_1, i_2)$ and $d_\mathcal{M}(S(i_1), S(i_2))$ for all $i_1, i_2 \in \mathcal{I}$ s.t. $i_1 \neq i_2$. This definition reflects an intuition that compositional languages should map similar (resp. dissimilar) inputs to similar (resp. dissimilar) messages. Following previous work using attribute-value objects (e.g., Chaabouni et al., 2020), we define $d_\mathcal{I}$ as the Hamming distance and $d_\mathcal{M}$ as the edit distance. Because this paper is about message segmentation, we can consider two types of edit distance. One is the "character" edit distance that regards elements $a \in \mathcal{A}$ as symbols. The other is the "word" edit distance that regards hypo-segments as symbols. Let us call the former *C-TopSim* and the latter *W-TopSim*.

## 4    PROBLEM DEFINITION

The purpose of this paper is to study whether Harris's articulation scheme (HAS) also holds in emergent languages. However, this question is too vague to answer. We first divide it into the following:

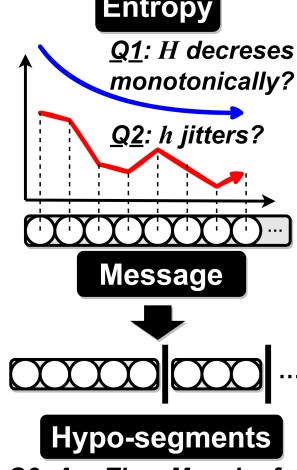

Q1. Does the conditional entropy $H$ decrease monotonically?

Q2. Does the branching entropy $h$ repeatedly fall and rise?

Q3. Do hypo-boundaries represent meaningful boundaries?

Q3 is the same as the original question, except that Q3 is slightly more formal. However, we have to answer Q1 and Q2 beforehand, because HAS implicitly takes it for granted that $H$ decreases monotonically and $h$ jitters. Although both Q1 and Q2 generally hold in natural languages, neither is trivial in emergent languages. Figure 2 illustrates Q1, Q2, and Q3.

Figure 2: Illustration of questions.

It is straightforward to answer Q1 and Q2 as we just need to calculate $H$ and $h$. In contrast, Q3 is still vague to answer, since we do not have prior knowledge about the boundaries of emergent languages and do not even know if they have such boundaries. To mitigate it, we posit the following necessary conditions for Q3. Let $G$ be a game $(\mathcal{D}_{n_{\text{val}}}^{n_{\text{att}}}, \mathcal{A}^k, S, R)$. If the answer to Q3 is yes, then:

C1. the mean number of hypo-boundaries per message should increase as $n_{\text{att}}$ increases,

C2. the size of the vocabulary (set of all hypo-segments) should increase as $n_{\text{val}}$ increases,

C3. W-TopSim should be higher than C-TopSim.

**About C1 and C2**    An attribute-value set $\mathcal{D}_{n_{\text{val}}}^{n_{\text{att}}}$ was originally introduced to measure compositionality. Compositionality, in this context, means how symbols in a message separately denote the components of meaning. In our case, each segment, or *word*, can be thought of as a certain unit that denotes the attribute values, so that the number of words in a message should increase as the corresponding attributes increase. Therefore, if the answer to Q3 is yes, then C1 should be valid. Likewise, the vocabulary size should be larger in proportion to the number of values $n_{\text{val}}$, motivating C2. Here, we mean by *vocabulary* the set of all hypo-segments. Note that the message length is fixed, because otherwise, the number of hypo-segments would be subject to variable message length as well as $(n_{\text{att}}, n_{\text{val}})$, and the implication of results would be obscure.

**About C3**    C3 comes from the analogy of the linguistic concept called *double articulation* (Martinet, 1960). In natural languages, meanings are quite arbitrarily related to the phonemes that construct them. In contrast, the meanings are less arbitrarily related to the words. The phonemes do not denote meaning units, but the words do. In our case, for example, the attribute-value object (RED, CIRCLE) seems less compositionally related to the character sequence "r,e,d,c,i,r,c,l,e", while it seems more compositionally related to the word sequence "red,circle." This intuition motivates C3.

Based on conditions C1, C2, and C3, Q3 is restated as follows: (Q3-1) Does the mean number of hypo-boundaries per message increase as $n_{\text{att}}$ increases?    (Q3-2) Does the vocabulary size increase

as $n_{\text{val}}$ increases? (Q3-3) Is W-TopSim higher than C-TopSim? Importantly, Q3-1, Q3-2, and Q3-3 are applicable to any word segmentation scheme as well as HAS, as they neither contain Eq. 1 nor Eq. 2 anymore. Thus, they can be general criteria for future work on the segmentation of emergent languages. We also verified that the above conditions C1, C2, and C3 are indeed satisfied by synthetic languages that are ideally compositional (see Appendix A).

# 5 EXPERIMENTAL SETUP

## 5.1 PARAMETER SETTINGS

**Input Space** $n_{\text{att}}$ and $n_{\text{val}}$ have to be varied to answer Q3-1, Q3-2, and Q3-3, while the sizes of the input spaces $|\mathcal{I}| = (n_{\text{val}})^{n_{\text{att}}}$ must be equal to each other to balance the complexities of games. Therefore, we fix $|\mathcal{I}| = 4096$ and vary $(n_{\text{att}}, n_{\text{val}})$ as follows:

$$(n_{\text{att}}, n_{\text{val}}) \in \{(1, 4096), (2, 64), (3, 6), (4, 8), (6, 4), (12, 2)\}. \tag{6}$$

**Message Space** The message length $k$ and alphabet $\mathcal{A}$ have to be determined for a message space $\mathcal{M} = \mathcal{A}^k$. We set $k = 32$, similarly to previous work on *ZLA* (Chaabouni et al., 2019; Rita et al., 2020; Ueda & Washio, 2021) that regards each $a \in \mathcal{A}$ as a "character." Note that $k = 32$ is set much longer than those of previous work on *compositionality* (Chaabouni et al., 2020; Ren et al., 2020; Li & Bowling, 2019) that typically adopts $k \doteq n_{\text{att}}$ as if each symbol $a \in \mathcal{A}$ were a "word." We set $\mathcal{A} = \{1, \ldots, 8\}$. Its size $|\mathcal{A}|$ should be as small as possible to avoid the data sparsity issue for boundary detection, and to ensure that each symbol $a \in \mathcal{A}$ serves as a "character." In preliminary experiments, we tested $|\mathcal{A}| \in \{2, 4, 8, 16\}$ and found that learning is stable when $|\mathcal{A}| \geq 8$.

**Architecture and Optimization** We follow Chaabouni et al. (2020) for agent architectures, an objective function, and an optimization method. The hidden size of GRU (Cho et al., 2014) is set to 500, following Chaabouni et al. (2020). All data from an input space $\mathcal{I} = \mathcal{D}_{n_{\text{val}}}^{n_{\text{att}}}$ are used for training. This dataset is upsampled to 100 times following the default setting of the code of Chaabouni et al. (2020). The learning rate is set to 0.001, which also follows Chaabouni et al. (2020). Based on our preliminary experiments to explore stable learning, a sender $S$ and a receiver $R$ are trained for 200 epochs, and the coefficient of the entropy regularizer is set to 0.01.

**Boundary Detection Algorithm** The boundary detection algorithm involves a parameter *threshold*. Since the appropriate value of *threshold* is unclear, we vary *threshold* as follows:

$$threshold \in \{0, 0.25, 0.5, 0.75, 1, 1.25, 1.5, 1.75, 2\}. \tag{7}$$

## 5.2 IMPLEMENTATION, NUMBER OF TRIALS, AND LANGUAGE VALIDITY

We implemented the code for training agents using the EGG toolkit (Kharitonov et al., 2019).[9] [10] EGG also includes the implementation code of Chaabouni et al. (2020), which we largely refer to. Each run took a few hours with a single GPU. In the following sections, an emergent language with a communication success rate of more than 90% is called *a successful language*.

# 6 RESULTS

As a result of training agents, we obtained 7, 8, 6, 8, 7, and 6 successful languages out of 8 runs for configurations $(n_{\text{att}}, n_{\text{val}}) = (1, 4096), (2, 64), (3, 16), (4, 8), (6, 4)$, and $(12, 2)$, respectively.

## 6.1 CONDITIONAL ENTROPY MONOTONICALLY DECREASES

To verify Q1, we show the conditional entropy $H(n)$ (Eq. 2) in Figure 3. In Figure 3, the conditional entropies of the successful languages (solid red lines) decrease monotonically. This confirms Q1 in successful languages. Interestingly, the conditional entropies of emergent languages derived

---

[9]The EGG toolkit: `https://github.com/facebookresearch/EGG`
[10]Our code: `https://github.com/wedddy0707/HarrisSegmentation`

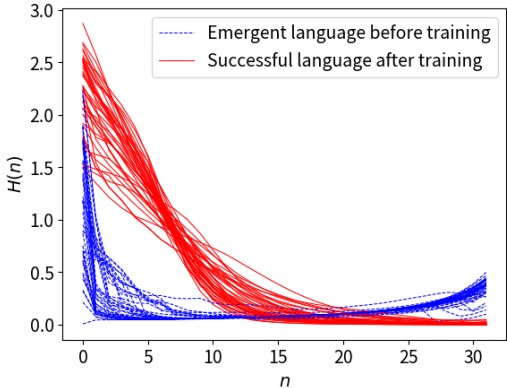
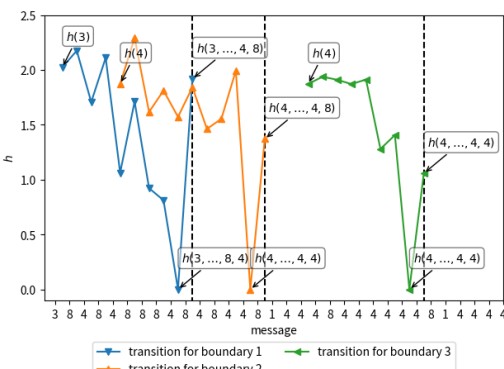

Figure 3: Conditional entropy $H(n)$. Dashed blue lines represent $H(n)$ of languages from untrained agents that finally learned successful languages, while solid red lines represent $H(n)$ of successful languages.

Figure 4: Example transition sequences of the branching entropy $h$ in a message "3,8,4,...,4,4,4" in a successful language for $(n_{\text{att}}, n_{\text{val}}) = (2, 64)$.

from untrained senders do not necessarily decrease, shown as dashed blue lines in Figure 3.[11] The monotonic decrease in conditional entropy emerges after training agents.

## 6.2 BRANCHING ENTROPY REPEATEDLY FALLS AND RISES

Next, to answer Q2, we computed the branching entropy $h(s)$ (Eq. 1) of the successful languages and applied boundary detection. As an example, we show a few actual transitions of $h(s)$ in Figure 4, in which the y-axis represents the value of $h(s)$ and the x-axis represents a message "3,8,4,...,4,4,4". The message is randomly sampled from a successful language when $(n_{\text{att}}, n_{\text{val}}) = (2, 64)$. The boundary detection algorithm with *threshold* $= 1$ yields three hypo-boundaries that are represented as dashed black lines in Figure 4. Blue, yellow, and green lines with triangle markers represent the transitions of $h(s)$ that yield hypo-boundaries. Note that the $(i + 1)$-th transition of $h(s)$ does not necessarily start from the $i$-th hypo-boundary, due to the definition of the algorithm. For instance, the second transition overlaps the first hypo-boundary. While the conditional entropy decreases monotonically as shown in Figure 3, the branching entropy repeatedly falls and rises in Figure 4. Moreover, we show the mean number of hypo-boundaries per message in Figure 5. Figure 5 indicates that for any $(n_{\text{att}}, n_{\text{val}})$ configuration, there are hypo-boundaries if *threshold* $< 2$, i.e., the branching entropy repeatedly falls and rises. These results validate Q2.

## 6.3 HYPO-BOUNDARIES MAY NOT BE MEANINGFUL BOUNDARIES

Next, we investigate whether Q3-1, Q3-2, and Q3-3 hold in successful languages. The results in the following sections falsify all of them. Thus, Q3 may not be true: hypo-boundaries may not be meaningful boundaries.

**Mean Number of Hypo-Boundaries per Message**   See Figure 5 again. The figure shows that the mean number of hypo-boundaries per message does not increase as $n_{\text{att}}$ increases. It does not decrease, either. This result falsifies Q3-1. Even when $n_{\text{att}} = 1$, there are as many hypo-boundaries as other configurations.

**Vocabulary Size**   Figure 6 shows the mean vocabulary sizes for each $(n_{\text{att}}, n_{\text{val}})$. The vocabulary size does not increase as $n_{\text{val}}$ increases, which falsifies Q3-2. However, focusing on $(n_{\text{att}}, n_{\text{val}}) \in \{(2, 64), (3, 16), (4, 8), (6, 4)\}$ and $0.25 \leq$ *threshold* $\leq 1$, there is a weak tendency to support C2. It suggests that hypo-segments are not completely meaningless either.

---

[11]One might think that the conditional entropy *cannot* increase by its definition. However, this is not the case in our setting (see Appendix B for more details).

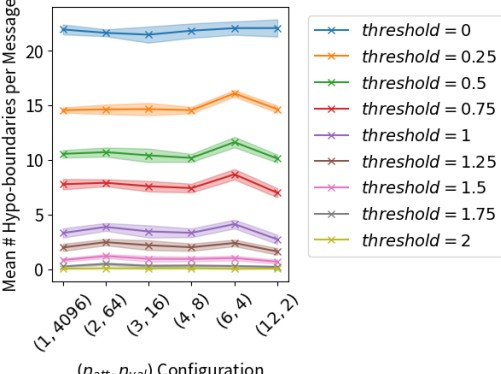

Figure 5: Mean number of hypo-boundaries per message in successful languages. *threshold* varies according to Eq. 7. Each data point is averaged over random seeds, and shaded regions represent one standard error of mean (SEM).

Figure 6: Vocabulary size in successful languages. *threshold* varies according to Eq. 7. Each data point is averaged over random seeds, and shaded regions represent one SEM.

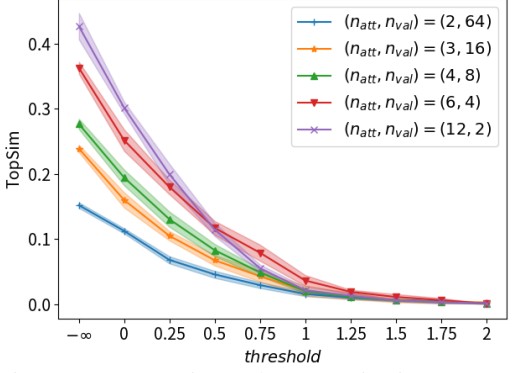

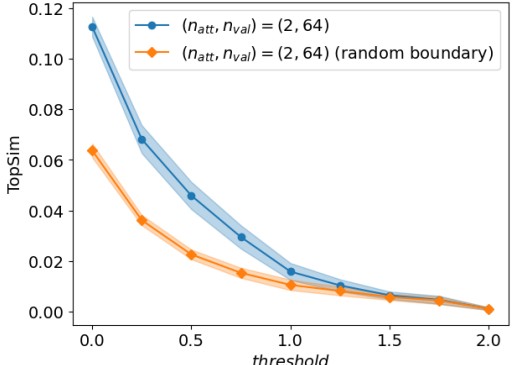

Figure 7: C-TopSim and W-TopSim in successful languages. "$-\infty$" corresponds to C-TopSim, while others correspond to W-TopSim. Each data point is averaged over random seeds, and shaded regions represent one SEM.

Figure 8: Hypo-boundary-based W-TopSim compared to random-boundary-based W-TopSim in successful languages for $(n_{\text{att}}, n_{\text{val}}) = (2, 64)$. Each data point is averaged over random seeds, and shaded regions represent one SEM.

**C-TopSim vs W-TopSim** Figure 7 shows C-Topsim and W-Topsim for each $(n_{\text{att}}, n_{\text{val}})$ and *threshold*.[12] Note that C-TopSim is TopSim with "character" edit distance, and W-TopSim is TopSim with "word" edit distance. In Figure 7, *threshold* $= -\infty$ corresponds to C-TopSim, while the others correspond to W-TopSim. [13] Our assumption in Q3-3 was C-TopSim < W-TopSim. On the contrary, Figure 7 shows a clear tendency for C-TopSim > W-TopSim, which falsifies Q3-3. Hypo-boundaries may not be meaningful. However, they may not be completely meaningless, either. This is because the hypo-boundary-based W-TopSim is higher than the random-boundary-based W-TopSim in Figure 8. Here, we mean by *random boundaries* the boundaries chosen randomly in the same number as hypo-boundaries in each message. Other $(n_{\text{att}}, n_{\text{val}})$ configurations show similar tendencies (see Appendix C).

## 6.4 FURTHER INVESTIGATION

**Ease-of-Teaching Setting Does Not Contribute to HAS** Methods proposed to improve TopSim might also contribute to giving rise to HAS. We picked up the *Ease-of-Teaching* (EoT, Li & Bowling, 2019), as it is known to be a simple and effective way to improve TopSim. However, the EoT setting did not contribute to the emergence of HAS, while improving C-TopSim (see Appendix D).

---

[12]Note that TopSim can only be defined when $n_{\text{att}} > 1$.

[13]If we were to apply boundary detection with *threshold* $= -\infty$, it would regard every data point in a message as a boundary. In other words, W-TopSim with *threshold* $= -\infty$ would be identical to C-TopSim. We adopt this notation in order to represent C-TopSim and W-TopSim in a unified manner in a single figure.

**Word Length and Word Frequency**  The results so far are related to *compositionality* of emergent languages (e.g., Kottur et al., 2017). We further associate our results with previous discussions on *Zipf's law of abbreviation* (ZLA) in emergent languages in Appendix E.

## 7 Discussion

In Section 6.1, we showed that the conditional entropy $H(n)$ decreases monotonically in emergent languages, confirming Q1. In Section 6.2, we demonstrated that the branching entropy $h(s)$ repeatedly falls and rises in emergent languages, which confirms Q2. It is an intriguing result, considering the discussions of Kharitonov et al. (2020), who showed that the entropy decreases to the minimum for successful communication if the message length $k = 1$. In contrast, our results suggest that the (branching) entropy does not simply fall to the minimum when the message length $k$ is longer. However, in Section 6.3, our results indicate that the hypo-boundaries may not be meaningful since Q3-1, Q3-2, and Q3-3 were falsified. Nevertheless, hypo-boundaries may not be completely meaningless either. This is because the hypo-boundary-based W-TopSim is higher than the random-boundary-based W-TopSim. It suggests that HAS-based boundary detection worked to some extent.

One limitation of this paper is that we focused on the specific neural architecture, i.e., GRU (Cho et al., 2014) as an agent basis, following Chaabouni et al. (2020). It remains to be answered whether articulation structure may arise in other architectures. Another limitation is that our HAS-based criteria highly relies on attribute-value settings, which obscures how to apply them to other settings, e.g., visual referential games (Havrylov & Titov, 2017; Choi et al., 2018). At least, however, C3 is applicable to the referential game, by using cosine dissimilarity between hidden states (Lazaridou et al., 2018). Finally, the question of why the current emergent languages fail to give rise to HAS is left open.

## 8 Related Work

**Segmentation of Emergent Language**  Resnick et al. (2020) proposed a compositionality measure called *residual entropy* (RE) that considers the *partition* of emergent messages, akin to the segmentation in this paper. However, RE is different from our HAS-based criteria in the following sense. First, RE assumes that all the messages are partitioned in the same way, i.e., boundary positions are the same in all the messages. Second, the computation of a partition involves an explicit reference to attribute values. This is opposed to HAS' statement that boundaries can be estimated without reference to meanings. Third, RE can be hard to compute.[14]

**Unsupervised Word Segmentation**  Regarding unsupervised word segmentation, this paper focused on HAS, for its simplicity, interpretability, and linguistic motivation. Of course, however, statistical models for word segmentation have made progress (e.g., Goldwater et al., 2009; Mochihashi et al., 2009). In addition, Taniguchi et al. (2016) tried to discover the double articulation structure (Martinet, 1960), i.e., both phonemes and words, from continuous speech signals. As emergent languages become more complex and natural-language-like in the future, it will be worthwhile to adopt more sophisticated segmentation methods.

## 9 Conclusion

This paper investigated whether Harris's articulation scheme (HAS) also holds in emergent languages. Emergent languages are artificial communication protocols emerging between agents. HAS has been considered a statistical universal in natural languages and can be used for unsupervised word segmentation. Our experimental results suggest that although emergent languages satisfy some prerequisites for HAS, HAS-based word boundaries may not be meaningful. Our contributions are (1) to focus on the word segmentation of emergent languages, (2) to pose verifiable questions to answer whether emergent languages have meaningful segments, and (3) to show another gap between emergent and natural languages. The standard signaling game satisfies prerequisites but still lacks some necessary ingredients for HAS.

---

[14]We have to compute entropy values for all the possible partitions to finally obtain RE. In our case, the number of possible partitions in one language can be astronomical up to $n_{\text{att}}{}^k = 12^{32} \approx 3.4 \times 10^{34}$.

ACKNOWLEDGMENTS

We would like to thank the anonymous reviewers for fruitful discussions and insightful comments.

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

# A    IDEALLY COMPOSITIONAL SYNTHETIC LANGUAGE INDEED SATISFIES CONDITIONS C1, C2, AND C3

The purpose of this supplemental section is to show that the HAS-based boundary detection algorithm works for an ideal language and that the ideal language indeed satisfies conditions C1, C2, and C3.

## A.1    SETUP

Recall that $n_{\text{att}}$ is the number of attributes, $n_{\text{val}}$ is the number of values, an attribute value set $\mathcal{D}^{n_{\text{att}}}_{n_{\text{val}}}$ is a meaning space, and $(n_{\text{att}}, n_{\text{val}}) \in \{(1, 4096), (2, 64), (3, 16), (4, 8), (6, 4), (12, 2)\}$. Set the alphabet size $|\mathcal{A}| = 8$ (same as the main content). Set the message length $k = 36$ (slightly different from the main content, but has a good property in the sense that $k \bmod n_{\text{att}} = 0$ for all $n_{\text{att}}$). Make an ideal synthetic language in the following procedure:

- Generate random segments $s_{a,v} = x^1_{a,v} \cdots x^{k/n_{\text{att}}}_{a,v}$ by $x^j_{a,v} \sim \text{Uniform}(\mathcal{A})$ with replacement for all $a \in \{1, \ldots, n_{\text{att}}\}$ and $v \in \{1, \ldots, n_{\text{val}}\}$. Assume that each $s_{a,v}$ denotes each attribute-value meaning unit: the value $v$ of the attribute $a$.
- Generate synthetic messages $m(v_1, \ldots, v_{n_{\text{att}}}) = s_{1,v_1} s_{2,v_2} \cdots s_{n_{\text{att}},v_{n_{\text{att}}}}$ by concatenating the segments $s_{j,v_j}$. Assume that each $m(v_1, \ldots, v_{n_{\text{att}}})$ denotes the corresponding attribute-value object $(v_1, \ldots, v_{n_{\text{val}}}) \in \mathcal{D}^{n_{\text{att}}}_{n_{\text{val}}}$.
- Create a synthetic language $L = \{m(v_1, \ldots, v_{n_{\text{att}}}) \mid (v_1, \ldots, v_{n_{\text{att}}}) \in \mathcal{D}^{n_{\text{att}}}_{n_{\text{val}}}\}$.

## A.2    RESULT

According to Table 1, All of C1, C2, and C3 are satisfied in the ideal synthetic languages. We also show the results of Table 1 in Figure 9, Figure 10, in the same manner as the main content.

| $n_{\text{att}}$ | $n_{\text{val}}$ | #boundaries | vocabulary size | C-TopSim | W-TopSim |
|---|---|---|---|---|---|
| 1 | 4096 | 1.52 ($\pm 0.01$) | 9407.1 ($\pm 37.8$) | (undefined) | (undefined) |
| 2 | 64 | 2.68 ($\pm 0.03$) | 394.8 ($\pm 7.9$) | 0.134 ($\pm 0.001$) | 0.465 ($\pm 0.010$) |
| 3 | 16 | 3.64 ($\pm 0.15$) | 96.8 ($\pm 3.2$) | 0.201 ($\pm 0.005$) | 0.465 ($\pm 0.010$) |
| 4 | 8 | 4.91 ($\pm 0.19$) | 62.2 ($\pm 2.6$) | 0.227 ($\pm 0.009$) | 0.627 ($\pm 0.021$) |
| 6 | 4 | 7.02 ($\pm 0.19$) | 39.7 ($\pm 1.3$) | 0.253 ($\pm 0.013$) | 0.763 ($\pm 0.015$) |
| 12 | 2 | 11.31 ($\pm 0.43$) | 28.2 ($\pm 0.8$) | 0.256 ($\pm 0.048$) | 0.862 ($\pm 0.009$) |

Table 1: The mean number of hypo-boundaries per message, the vocabulary size, C-TopSim, and W-TopSim of ideal synthetic languages. We obtained eight languages for each $(n_{\text{att}}, n_{\text{val}})$ with different random seeds. ($\pm \cdot$) represents the standard error.

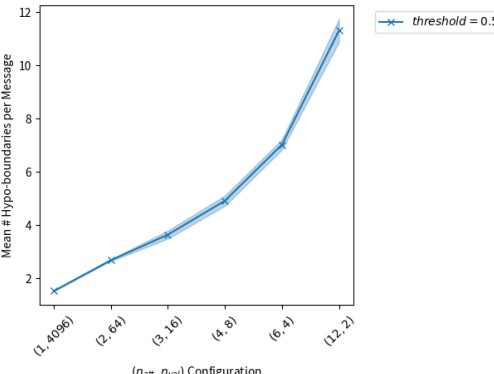 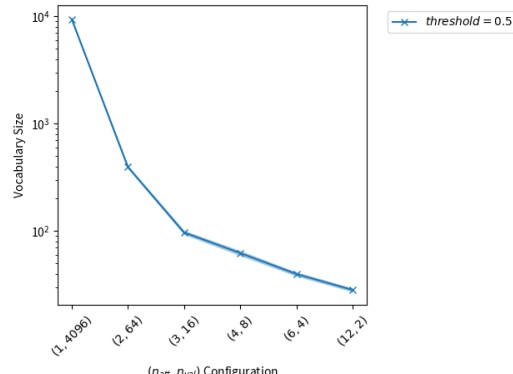

Figure 9: Mean number of hypo-boundaries per message in synthetic languages. Each data point is averaged over random seeds, and shaded regions represent one standard error of mean (SEM).

Figure 10: Vocabulary size in synthetic languages. Each data point is averaged over random seeds, and shaded regions represent one SEM. The y-axis is on the log scale, as the datapoint for $(n_{\mathrm{att}}, n_{\mathrm{val}}) = (1, 4096)$ is very large compared to the other datapoints.

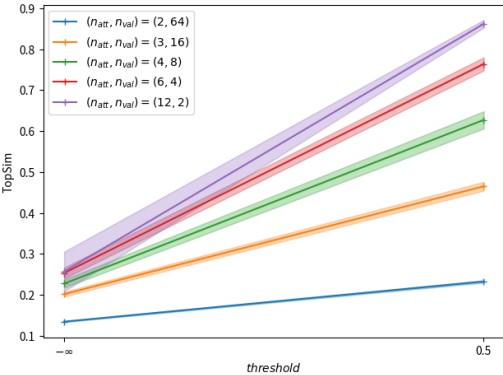

Figure 11: C-TopSim and W-TopSim in synthetic languages. "$-\infty$" corresponds to C-TopSim, while other *threshold* values correspond to W-TopSim. Each data point is averaged over random seeds, and shaded regions represent one SEM.

## B  WHY CAN CONDITIONAL ENTROPY INCREASE IN SIGNALING GAME?

One might wonder why the conditional entropy $H(n)$ can increase and think it cannot due to its definition. This is true when we have a single (possibly infinite) sequence. For example, the conditional entropy of an infinite monkey typing sequence is constant since for any $n \in \mathbb{N}$ and $s \in \mathcal{X}^n$,

$$h(s) = -\sum_{x \in \mathcal{X}} P(x \mid s) \log_2 P(x \mid s) = -\sum_{x \in \mathcal{X}} |\mathcal{X}|^{-1} \log_2 |\mathcal{X}|^{-1} = \log_2 |\mathcal{X}|,$$

$$H(n) = \sum_{s \in \mathcal{X}^n} P(s)h(s) = \log_2 |\mathcal{X}|.$$

In general, the conditional entropy $H(n)$ of a single sequence is a weakly decreasing function. However, emergent languages arising from signaling games are not single sequences. Each emergent language is a set of finite sequences: $L = \{S(i) \in \mathcal{M} \mid i \in \mathcal{I}\}$. Consider, for instance, the following toy language:

$$L_{\text{toy}} = \left\{ \begin{array}{c} aaaaa \\ aaaab \\ aaaac \end{array} \right\}.$$

In $L_{\text{toy}}$, $H(1) < H(4)$ holds, as a symbol after a unigram is most likely to be $a$, while a symbol after a 4-gram is equally likely to be $a$, $b$, and $c$.

## C  HYPO-BOUNDARY-BASED W-TOPSIM AND RANDOM-BOUNDARY-BASED W-TOPSIM

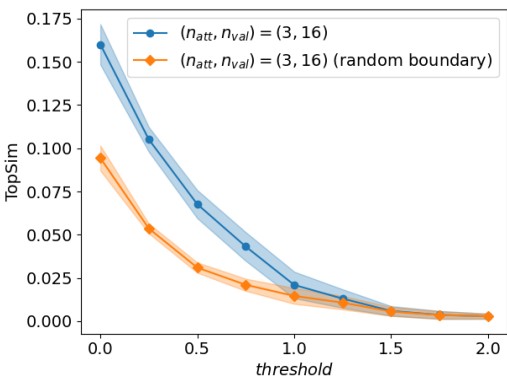

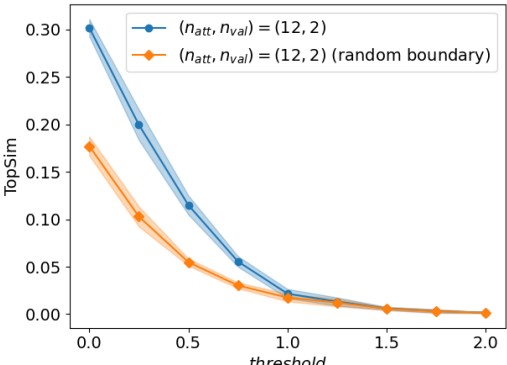

Figure 12: Hypo-boundary-based W-TopSim compared to random-boundary-based W-TopSim in successful languages for $(n_{\text{att}}, n_{\text{val}}) = (3, 16)$. Each data point is averaged over random seeds, and shaded regions represent one SEM.

Figure 13: Hypo-boundary-based W-TopSim compared to random-boundary-based W-TopSim in successful languages for $(n_{\text{att}}, n_{\text{val}}) = (4, 8)$. Each data point is averaged over random seeds, and shaded regions represent one SEM.

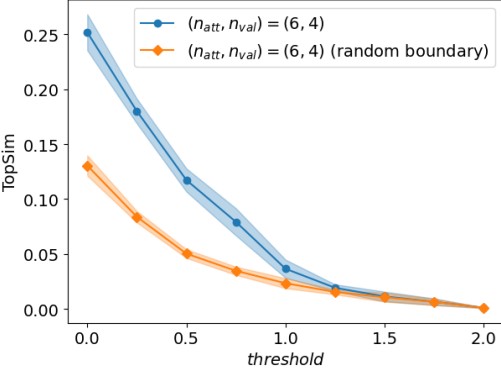

Figure 14: Hypo-boundary-based W-TopSim compared to random-boundary-based W-TopSim in successful languages for $(n_{\text{att}}, n_{\text{val}}) = (6, 4)$. Each data point is averaged over random seeds, and shaded regions represent one SEM.

Figure 15: Hypo-boundary-based W-TopSim compared to random-boundary-based W-TopSim in successful languages for $(n_{\text{att}}, n_{\text{val}}) = (12, 2)$. Each data point is averaged over random seeds, and shaded regions represent one SEM.

## D  EASE-OF-TEACHING SETTING DOES NOT CONTRIBUTE TO HAS

Some methods proposed to improve TopSim might also contribute to the emergence of HAS. We picked up the *Ease-of-Teaching* (EoT) setting (Li & Bowling, 2019), as it is known to be a simple and effective way to improve TopSim. As a result, the EoT setting did not contribute to the emergence of HAS, while improving TopSim (C-TopSim).

**Setup**  The EoT setting is quite simple: it resets the receiver agent periodically during training time. In this paper, we reset the receiver every 20 epochs.

**Result**  We show the results in Figure 16, Figure 17, and Figure 18. The mean number of hypo-boundaries does not increase as $n_{att}$ increases in Figure 16. The vocabulary size does not increase as $n_{val}$ increases in Figure 17. C-TopSim > W-TopSim in Figure 18.

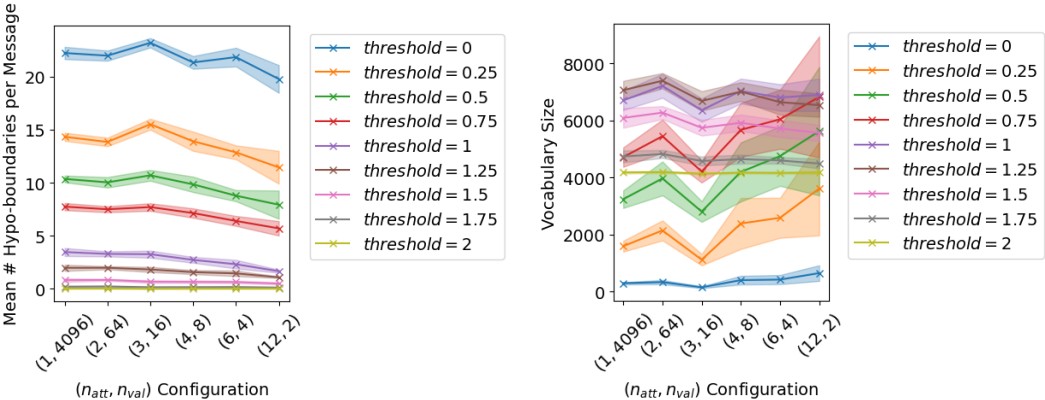

Figure 16: Mean number of hypo-boundaries per message in successful languages with the EoT setting. Each data point is averaged over random seeds, and shaded regions represent one SEM.

Figure 17: Vocabulary size in successful languages with the EoT setting. Each data point is averaged over random seeds, and shaded regions represent one SEM.

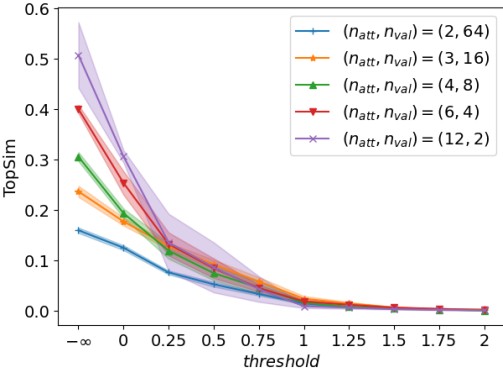

Figure 18: C-TopSim and W-TopSim in successful languages with the EoT setting. "$-\infty$" corresponds to C-TopSim, while other *threshold* values correspond to W-TopSim. Each data point is averaged over random seeds, and shaded regions represent one SEM.

# E  HYPO-SEGMENTS AND ZIPF'S LAW OF ABBREVIATION

The results in the main content are related to *compositionality* of emergent languages (e.g., Kottur et al., 2017). In this supplemental section, we further associate our results with previous discussions on *Zipf's law of abbreviation* (ZLA) in emergent languages (Chaabouni et al., 2019; Rita et al., 2020; Ueda & Washio, 2021). ZLA is known as a statistical property in natural languages that the more frequently a word is used, the shorter it is (Zipf, 1935). By considering hypo-segments as "words," we can check whether hypo-segments follow ZLA. Figure 19 shows the hypo-segment lengths sorted by frequency rank for $(n_{\text{att}}, n_{\text{val}}) = (1, 4096)$.[15] If hypo-segments follow ZLA ideally, they should show a monotonic increase. The distribution of the lengths of the hypo-segments shows a clear ZLA-like tendency for *threshold* $\in \{0, 0.5\}$, although the tendencies are less clear for the other *threshold*.[16] It means that hypo-segments follow ZLA with an appropriate *threshold* value. Other $(n_{\text{att}}, n_{\text{val}})$ configurations show similar tendencies (see Figure 20, Figure 21, Figure 22, Figure 23, and Figure 24).

This is a suggestive result because we neither imposed a length penalty on messages (Chaabouni et al., 2020), modeled the laziness/impatience of agents (Rita et al., 2020), nor modeled short-term memories (Ueda & Washio, 2021). Note that it may be just an artifact, analogous to the fact that even a monkey typing sequence divided by the "white space" follows ZLA (Miller, 1957).

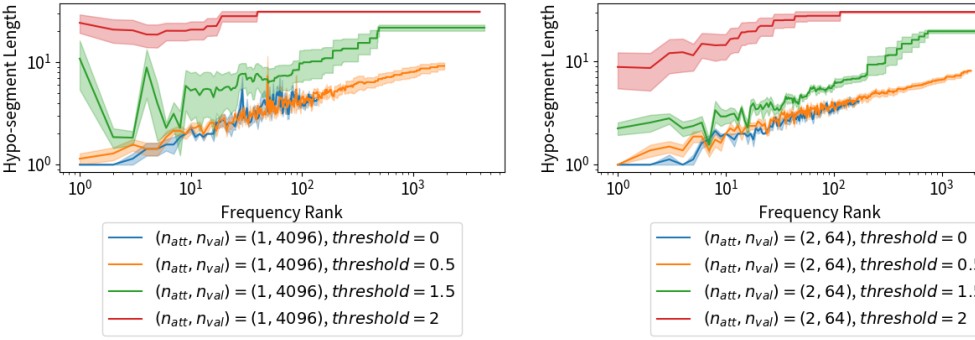

Figure 19: Hypo-segment lengths sorted by frequency rank for $(n_{\text{att}}, n_{\text{val}}) = (1, 4096)$. Each data point is averaged over random seeds, and shaded regions represent one SEM.

Figure 20: Hypo-segment lengths sorted by frequency rank for $(n_{\text{att}}, n_{\text{val}}) = (2, 64)$. Each data point is averaged over random seeds and shaded regions represent one SEM.

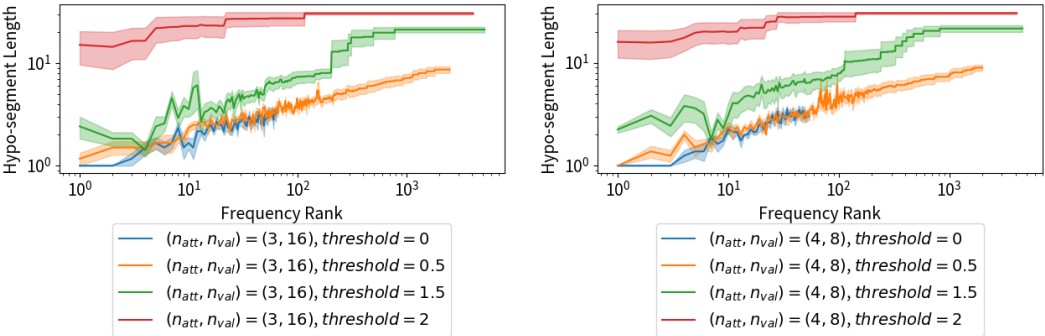

Figure 21: Hypo-segment lengths sorted by frequency rank for $(n_{\text{att}}, n_{\text{val}}) = (3, 16)$. Each data point is averaged over random seeds and shaded regions represent one SEM.

Figure 22: Hypo-segment lengths sorted by frequency rank for $(n_{\text{att}}, n_{\text{val}}) = (4, 8)$. Each data point is averaged over random seeds and shaded regions represent one SEM.

---

[15]We picked up only *threshold* $\in \{0.5, 1.5, 2\}$ and adopted a log-log graph for readability.

[16]The plot shows the zigzagging behavior for *threshold* = 1.5, and most of the hypo-segment lengths hit the message length $k = 32$ for *threshold* = 2.

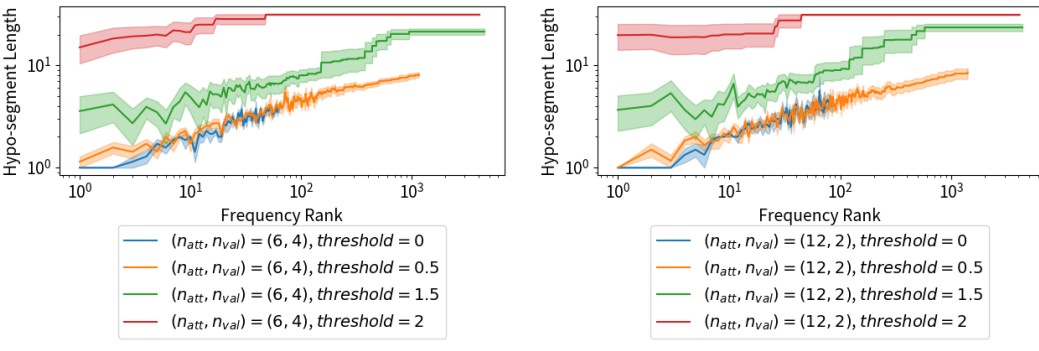

Figure 23: Hypo-segment lengths sorted by frequency rank for $(n_{\text{att}}, n_{\text{val}}) = (6, 4)$. Each data point is averaged over random seeds and shaded regions represent one SEM.

Figure 24: Hypo-segment lengths sorted by frequency rank for $(n_{\text{att}}, n_{\text{val}}) = (12, 2)$. Each data point is averaged over random seeds and shaded regions represent one SEM.

## F  ILLUSTRATION OF BOUNDARY DETECTION ALGORITHM

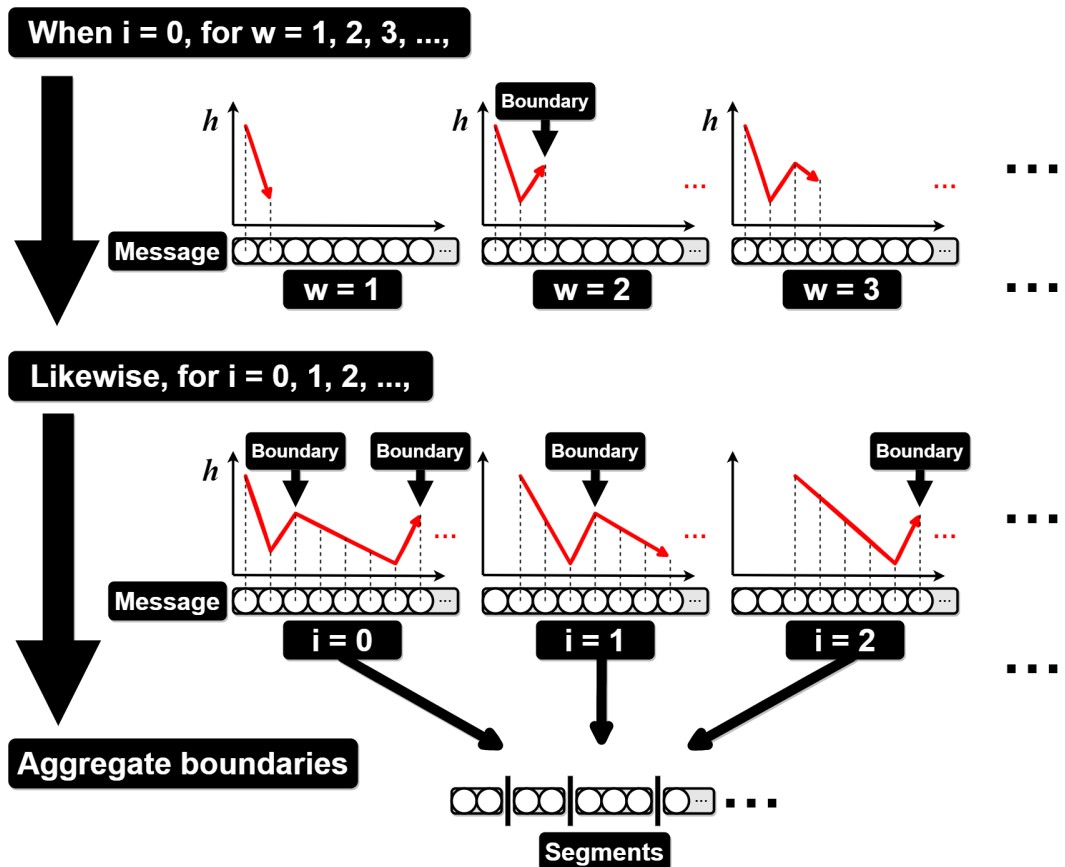

Figure 25: Illustration of HAS-based boundary detection algorithm (Algorithm 1).

