# OpenReview forum: "On the Word Boundaries of Emergent Languages Based on Harris's Articulation Scheme"
_ICLR.cc/2023/Conference — ICLR 2023 poster_

### Official Review · Reviewer_XTS9 · 2022-10-24

**Confidence:** 3
**Correctness:** 4
**Technical Novelty And Significance:** 3
**Empirical Novelty And Significance:** 3
**Recommendation:** 6

**Clarity, Quality, Novelty And Reproducibility:**

Quality:

As mentioned above, this is careful work. I see no problems with the math, the notation, the experimental design or the results.

Clarity:

The paper is dense but ultimately easy enough to follow. I found one typo: in the first sentence of the introduction, “HAS-baseThisboundary detection algorithm.”

One change that could improve clarity would be, for the assessment of synthetic languages in Appendix A, to present the data using the same Figures used to assess C1-C3 (Figures 5-8) in Section 6.3. This way we could contrast what a successful experiment results would look like against a negative result.

Originality:

In terms of technical contribution, this is a simple application of a known unsupervised word segmentation technique to a known problem. That combination is novel, though, and the method is relatively obscure and this paper serves as an excellent re-introduction to it. As mentioned above, we should also consider the assessment criteria C1-C3 to be novel and useful when assessing meaningful word formation in emergent languages.

**Strength And Weaknesses:**

Strengths:

This is a very careful, thorough paper. Everything is carefully defined and described. It is dense, but ultimately easy to follow.

This paper has two paths to influence: one by reminding the community of this method and applying it to emergent languages, and one by defining criteria that can be used to assess the meaningfulness of word-like units discovered in emergent languages.

The inclusion of an assessment of a synthetic language in Appendix A is a really nice touch, and greatly strengthens the paper, providing confidence that the conditions do indeed work for languages that verifiably have meaningful words.

Weaknesses:

It is unclear to me how important word formation is in current studies of emergent languages. As the authors themselves point out, most previous work implicitly assumes one-character-per-word.

As this appears to be the first paper to dive into word formation, it would be interesting to talk about the ramifications of some of the changes they’ve made to the set-up. Did forcing multiple-character representations reduce the success rate or the scores of successful systems? Are there any downsides to using fixed-length messages when hoping for meaningful words? I.E. Couldn’t the system then assign meanings to certain characters in certain positions? The paper mentions only experimental benefits of fixed-length messages.

The work is very specific to this particular technique. It has the feel of an author dusting off a favorite tool and aiming it at a new problem. I worry that this will cap the amount of interest this paper can draw from the emergent language community. In my review, I have tried to emphasize the portions of the paper that apply to any segmentation scheme. I think it would benefit the paper for the authors to do the same.


**Summary Of The Paper:**

This paper proposes a method for doing word segmentation in emergent languages derived during a one-way signaling game. The method is based on Harris’s articulation scheme, and involves tracking a branching entropy term for a rapid rise. The method is light-weight and can be derived from simple surface (n-gram) statistics, and has been used in a series of previous word segmentation papers. Perhaps more importantly, the authors propose a set of criteria (C1-C3 in Section 4) to determine whether meaningful word-like units have been discovered. This is nothing specific to the Harris articulation scheme in these criteria, and they could likely be used to assess other segmentation methods or other communication-game learning methods for meaningful words. The conclusion of evaluating the proposed method with these criteria is that the preconditions for this method have been met, but there is as of yet no evidence that meaningful units have been discovered.

**Summary Of The Review:**

This paper feels a little niche, both in that I do not have a strong notion whether the word formation problem is an important one in emergent languages at this point, nor do I know whether this proposed technique is the best approach for finding word boundaries if they do exist. Regardless of the nicheness of the problem, the paper is well-written, and builds a strong evaluation framework for other groups to use, validating that framework with a synthetic language in Appendix A. The authors use this framework to draw the conclusion that for this configuration of the signaling game, no meaningful units are emerging, which definitely raises some interesting questions.

---

> ### Comment · Reviewer_BukJ · 2022-11-14
> **Motivation for emergent language community**
>
> > It is unclear to me how important word formation is in current studies of
> > emergent languages. As the authors themselves point out, most previous work
> > implicitly assumes one-character-per-word.
>
> I see this is an important step since the one-character-per-word assumptions
> seems to be a sort of default position without explicit justification.
>
> > The work is very specific to this particular technique. It has the feel of an
> > author dusting off a favorite tool and aiming it at a new problem. I worry
> > that this will cap the amount of interest this paper can draw from the
> > emergent language community. In my review, I have tried to emphasize the
> > portions of the paper that apply to any segmentation scheme. I think it would
> > benefit the paper for the authors to do the same.
>
> I am not familiar (breadth- or depth-wise) with segmentation schemes, so
> I cannot make judgment no to what degree HAS is or is not unique here.  What
> I can say, though, is that segmentation is not frequently addressed in emergent
> language despite its relevance.  Furthermore, the fact that the segmentation
> scheme has some basis in linguistics (as opposed to some engineered NLP
> solution), which would naturally facilitate comparison to human language,
> further bolstering the relevance of the paper.  Even if this just a dusted-off
> technique, I think it is still informative to the emergent language community
> which skews heavily towards researchers with a background in multi-agent RL and
> deep learning, not linguistics.
>
> > This paper feels a little niche, both in that I do not have a strong notion
> > whether the word formation problem is an important one in emergent languages
> > at this point
>
> More or less addressed above, so I will simply state that the nature of "words"
> in emergent language is relevant and not often studied.

---

> ### Author Response · Authors · 2022-11-18
> **First Response to Reviewer XTS9**
>
> Thank you very much for reviewing our paper and indicating our contribution from another angle, which is somewhat unclear in the original version of our paper.
>
> > It is unclear to me how important word formation is in current studies of emergent languages. As the authors themselves point out, most previous work implicitly assumes one-character-per-word.
>
> Please see the first reply to Reviewer zwe8.
>
> > As this appears to be the first paper to dive into word formation, it would be interesting to talk about the ramifications of some of the changes they’ve made to the set-up. Did forcing multiple-character representations reduce the success rate or the scores of successful systems?
>
> [Just to confirm] You mean you are interested in e.g., the relationship between training accuracy and (message_length, vocabulary_size) configuration.
>
> Although we did not include them for it seems to be out of our paper’s scope, it might be interesting to show such results (in the appendix of the final version).
>
> > Are there any downsides to using fixed-length messages when hoping for meaningful words? I.E. Couldn’t the system then assign meanings to certain characters in certain positions? The paper mentions only experimental benefits of fixed-length messages.
>
> The fixed-length setting may be a bit odd, considering the fact that the word length varies in natural language, though we just follow the standard way and most emecom work has the same issue.
>
> > The work is very specific to this particular technique. It has the feel of an author dusting off a favorite tool and aiming it at a new problem. I worry that this will cap the amount of interest this paper can draw from the emergent language community. In my review, I have tried to emphasize the portions of the paper that apply to any segmentation scheme. I think it would benefit the paper for the authors to do the same.
>
> [Just to confirm]
> - For the former part: you mean you are concerned about the appropriateness of using a particular technique like HAS.
> - For the latter part: you mean that C1, C2, and C3 are applicable to any segmentation methods as well as HAS, so we should emphasize it as another contribution.
>
> [For the former part]
>
> The use of HAS can be justified. See the last reply to Reviewer nRa3.
>
> [For the latter part]
>
> That is a great idea! In the revised version, we inserted the following sentence in the 3rd paragraph, Section 1 (Introduction):
> - Importantly, some of the questions are applicable to any word segmentation scheme as well as HAS, so that they can be general criteria for future work on the segmentation of emergent languages.
>
> and the following sentences at the bottom of Section 4 (Problem Definition):
> - Importantly, Q3-1, Q3-2, and Q3-3 are applicable to any word segmentation scheme as well as HAS, as they neither contain Eq. 1 nor Eq. 2 anymore. Thus, they can be general criteria for future work on the segmentation of emergent languages.
>
> > One change that could improve clarity would be, for the assessment of synthetic languages in Appendix A, to present the data using the same Figures used to assess C1-C3 (Figures 5-8) in Section 6.3. This way we could contrast what a successful experiment results would look like against a negative result.
>
> Thank you for pointing this out!
> In the revised version, we added the figure versions.

---

### Official Review · Reviewer_PmT4 · 2022-10-24

**Confidence:** 5
**Correctness:** 2
**Technical Novelty And Significance:** 2
**Empirical Novelty And Significance:** 3
**Recommendation:** 5

**Clarity, Quality, Novelty And Reproducibility:**

The main contribution is about the Boundary Detection algorithm which makes use of conditional entropy to check for word boundaries. As I recall, one previous work [1] has done a similar study on using residual entropy for evaluating compositionality in emergent languages. There are no comparisons to this metric and how the proposed method add anything novel on top of this work.

Besides does the hypothesis still hold when using visual / multimodal inputs [2] that propose different factors for compositional emergent languages?

[1] Gupta et al. 2020. Compositionality and Capacity in Emergent Languages.

[2] Choi et al. 2018. Compositional obverter communication learning from raw visual input.


**Strength And Weaknesses:**

The paper proposes a new research direction/benchmark for emergent languages to be as close to natural language. The direction is indeed interesting and important for the research on emergent languages as the main goal is to achieve human interpretable languages.

However, the analysis done in the paper is more quite limited to come to a definite conclusion. The only environment chosen to perform the tests is the reconstruction game which basically just emulates the encoder-decoder setup. I would encourage the authors to carry out similar tests for referential games and other derived games that have been used in emergent communication literature.

**Summary Of The Paper:**

The paper investigates word boundaries in the context of emergent languages. The hypothesis is that if we want emergent languages to follow properties of natural languages like compositionality, they should have meaningful word boundaries as defined by Harris Articulation Scheme. The authors propose derived tests to check whether a given emergent language possess this property. They show that factors encouraging compositionality in emergent languages do not correlate with languages having this property.

**Summary Of The Review:**

The paper is still a work in progress and I would encourage the authors to bolster their claims by adding diverse experiments and a more thorough literature survey adding comparisons with prior work.

---

> ### Comment · Reviewer_BukJ · 2022-11-14
> **Comparing reviews**
>
> > However, the analysis done in the paper is more quite limited to come to
> > a definite conclusion. The only environment chosen to perform the tests is
> > the reconstruction game which basically just emulates the encoder-decoder
> > setup. I would encourage the authors to carry out similar tests for
> > referential games and other derived games that have been used in emergent
> > communication literature.
>
> This is a limitation for most emergent language papers which (1) introduce
> something new, (2) provide a robust analysis of results, and (3) fit within
> a conference page limit.  This certainly limits the empirical impact of this
> paper, but I believe the empirical contributions are still sufficient to
> motivate the interest of the theoretical contributions, which I hold to be more
> important.
>
> > The main contribution is about the Boundary Detection algorithm which makes
> > use of conditional entropy to check for word boundaries. As I recall, one
> > previous work [1] has done a similar study on using residual entropy for
> > evaluating compositionality in emergent languages. There are no comparisons
> > to this metric and how the proposed method add anything novel on top of this
> > work.
>
> Similarly to this paper, the residual entropy metric introduced in [1] does
> look at alternative segmentations for studying compositionality in emergent
> language.  Nevertheless, the investigation of and with this metric are very
> brief in [1] -- it does not provide any further analysis of the structure of
> the segmentations.  This paper, on the other hand, gives a more robust
> investigation, explanation, and background segmentation proposed.  I definitely
> agree, that a mention and at least brief comparison would be beneficial, but
> the depth of this paper sufficiently distinguishes it from [1].

---

> ### Author Response · Authors · 2022-11-18
> **First Response to Reviewer PmT4**
>
> Thank you very much for reviewing our paper. We think that some of your points are crucial for improving our paper.
>
> > As I recall, one previous work [1] has done a similar study on using residual entropy for evaluating compositionality in emergent languages. There are no comparisons to this metric and how the proposed method add anything novel on top of this work.
>
> Thank you for sharing this information! We think that it is worth mentioning in the paper.
> In the revised version, we made the Related Work section and mention the residual entropy.
>
> Nevertheless, the residual entropy is different from our HAS-based criteria in the following sense, and thus the novelty of our study still holds.
>
> 1. *the residual entropy assumes that all the messages are partitioned in the same way*, i.e., boundary positions are the same in all the messages. This is different from the HAS-based algorithm.
> 2. *the computation of a partition involves an explicit reference to attribute values*. This is opposed to HAS’ statement that “boundaries can be estimated to some extent without reference to meanings in natural languages.”
> 3. *the residual entropy can be hard to compute*. We have to compute the entropy score for all the possible partitions to finally obtain the residual entropy. In our case, for example, the number of possible partitions in one language can be astronomical up to $(n_{\rm att})^k=12^{32}\approx 3.4\times 10^{34}$.
>
> Due to the computational difficulty, we could not conduct additional experiments for comparison between the residual entropy and HAS.
>
> P.S. [1] seems a workshop paper. Instead, we found and cited its conference version.
>
> > Besides does the hypothesis still hold when using visual / multimodal inputs [2] that propose different factors for compositional emergent languages?
>
> Most emecom papers that employ Lewis’s signaling game or its variant employ either an attribute-value-based signaling game or visual referential game and make the best use of each. We believe that the fact that our paper does not use a visual referential game will not be a reason to reject it, though it is a limitation. In the revised version, we mention it as a limitation in the Discussion section (Section 7).

---

> > ### Comment · Reviewer_PmT4 · 2022-12-01
> > **Rebuttal response**
> >
> > I appreciate the authors for clarifying concerns related to prior work. I now agree that the comparison, albeit necessary, shows the contrast between the two metrics. I have updated my score accordingly.
> >
> > Although, I am skeptical that results purely based on a Lewis signaling game could lead to the rejection of the alternate hypothesis.

---

### Official Review · Reviewer_nRa3 · 2022-10-25

**Confidence:** 3
**Correctness:** 4
**Technical Novelty And Significance:** 3
**Empirical Novelty And Significance:** 3
**Recommendation:** 8

**Clarity, Quality, Novelty And Reproducibility:**

The paper's quality and clarity are satisfactory and provide interesting insight. The finding is worth sharing in the community of emergent communication. Reproducibility is also acceptable. However, the novelty is questionable.



**Strength And Weaknesses:**

Strength
+ The paper clearly shows that articulation structure, HAS, does not emerge in the sense of entropy-based measure in a type of emergent communication
+ The paper provides an interesting suggestion about the limitation of the current emergent communication model.

Weakness
- The paper did not discover the reason why the current emergent communication model does not make HAS emerge.
- Apparently, the property depends on the implicit definition of language models that are used in emergent communication models. This analysis is just about a type of emergent communication model and architecture.



Also, regarding word segmentation, the authors have mentioned HAS and Tanaka-Ishii studies. However, statistical models for word segmentations have made progress. The author refers to such studies. Also, even simultaneous phone and word discovery statistical models have been proposed.

[Word discovery]
[1] S. Goldwater, T. L. Griffiths, and M. Johnson, "A Bayesian framework for word segmentation: Exploring the effects of context," Cognition, vol. 112, no. 1, pp. 21–54, Jul. 2009.
[2] D. Mochihashi, T. Yamada, and N. Ueda, "Bayesian unsupervised word segmentation with nested Pitman-Yor language modeling," in Proc. Joint Conf. 47th Annu. Meeting ACL 4th Int. Joint Conf. Nat. Lang. Process. AFNLP (ACL-IJCNLP), Singapore, 2009, pp. 100–108

[Phone and word discovery]
[3] Tadahiro Taniguchi, Shogo Nagasaka, Ryo Nakashima, Nonparametric Bayesian Double Articulation Analyzer for Direct Language Acquisition from Continuous Speech Signals, IEEE Transactions on Cognitive and Developmental Systems, Vol.8 (3), pp. 171-185 .(2016) DOI: 10.1109/TCDS.2016.2550591 (Open Access)

You may be interested in word segmentation with co-occuerence cues in relation to "meaningful" segments.
[4] Tomoaki Nakamura, Takayuki Nagai, Kotaro Funakoshi, Shogo Nagasaka, Tadahiro Taniguchi and Naoto Iwahashi, Mutual Learning of an Object Concept and Language Model Based on MLDA and NPYLM, 2014 IEEE/RSJ International Conference on Intelligent Robots and Systems (IROS'14), 2014

As mentioned above, the phenomenon of the emergence of word segments depends on the implicitly designed language model, i.e., GRU in this model. Therefore, it is essential to clarify this point in the discussion of the general nature of the possibility of the emergence of articulation structure.


**Summary Of The Paper:**

This paper analyzes the articulation structure that could emerge in emergent language. The paper follows the studies of deep learning-based emergent communication models, which have been studied for half of the decade. In particular, the study adopts the framework of Chaabouni et al. (2020).
The authors tested if they satisfied HAS. HAS states that word boundaries can be obtained solely from phonemes in natural language. We adopted  HAS-based word segmentation and verified whether emergent languages have meaningful word segments. The experiment suggested they do not have. They found that the current emergent language model is still missing some necessary ingredients.


**Summary Of The Review:**


This paper analyzes the articulation structure that could emerge in emergent language. The paper clearly shows that articulation structure, HAS, does not emerge in the sense of entropy-based measure in a type of emergent communication. However, because of the lack of theoretical insight of the architecture of emergent communication, the value of the obtained insight is limited to the model they adopted. It means this is a kind of "case study." As a result, I am not sure if the paper contains a sufficient contribution to appear as an ICLR paper.


[Update] Considering the authors' revision, I have updated my score.

---

> ### Author Response · Authors · 2022-11-18
> **First Response to Reviewer nRa3**
>
> Thank you very much for reviewing our paper. The information you shared is really helpful.
>
> > The paper did not discover the reason why the current emergent communication model does not make HAS emerge.
>
> Thank you for pointing this out.
> We have to recognize it as a limitation.
> In the revised version, we mention it in the Discussion section (Section 7).
>
> > Apparently, the property depends on the implicit definition of language models that are used in emergent communication models. This analysis is just about a type of emergent communication model and architecture.
>
> We agree with it, too.
> In the revised version, we mention it in the Discussion section (Section 7).
>
> > Also, regarding word segmentation, the authors have mentioned HAS and Tanaka-Ishii studies. However, statistical models for word segmentations have made progress. The author refers to such studies. Also, even simultaneous phone and word discovery statistical models have been proposed.
>
> Thank you very much for sharing useful information!
> In the revised version, we made the Related Work section (Section 8) and mention them as related work.
>
> Nevertheless, the use of HAS can be justified as follows:
>
> Other advanced methods try to consider both relationships within words (i.e., character n-grams) and between words (i.e., word n-grams) to achieve better F scores for natural language corpora.
> However, such "everything-taken-into-consideration" methods may obscure the implication of results in the emergent communication study, because we have no prior knowledge on emergent language, no ground-truth data of segmentation, and it is not even evident, in the first place, whether they have meaningful segments.
> On the other hand, HAS is based on linguistic motivation and focuses solely on relationships within words (i.e., character n-grams). That is, our paper found that the current emergent languages lack natural-language-like articulation structure, in particular, they lack proper relationships within words.

---

> > ### Comment · Reviewer_nRa3 · 2022-11-23
> > **Thanks for your revision.**
> >
> > Thank you for the revision. I believe this revision has made the paper more significant.

---

### Official Review · Reviewer_zwe8 · 2022-10-25

**Confidence:** 4
**Correctness:** 3
**Technical Novelty And Significance:** 3
**Empirical Novelty And Significance:** 3
**Recommendation:** 5

**Clarity, Quality, Novelty And Reproducibility:**

* Clarity: I find that the text is generally clear;
* Novelty: I believe the topic of the study is largely novel, however, I would appreciate if authors could better differentiate it from the body of research on compositionality of the emergent languages;
* Reproducibility: while the authors do not mention if the code for the experiments is going to be released, it is largely based on a public codebase.


**Strength And Weaknesses:**

Strengths:
* I believe the paper brings an interesting toolset to the topic of analyzing emergent languages
* The experimental study is convincing

Weaknesses:
* Motivation: We already have a strong understanding that human language-like, interpretable features (ZLA, compositionality) often do not emerge by themselves in emergent languages. For now, looking at HAS is motivated as yet “another direction” to study emergent languages. I think the paper could benefit from highlighting why it is particularly worth studying. Furthermore, If HAS is important, can we take the findings of the paper and find a way to enforce it?
* In my view, the search for “word” boundaries seems to be closely related to the search of compositionality, i.e. whether there are elementary units within messages with a fixed “meaning”  (Chaabouni et al., 2020). I believe the text could benefit from contrasting those pursuits. For instance, if compositionality in the sense of (Chaabouni et al., 2020) is achieved, would word boundaries become meaningful? or vice-versa?
* Further, as formulated, the word boundaries segment the messages into groups of adjacent tokens (ie. the 2nd token cannot be grouped with the 4th token w/o being grouped with the 3rd token). Is there an intuition why this should hold for the artificial agents? As far as I understand, any successful language can be modified by a fixed permutation of positions w/o sacrificing its expressiveness. For instance, the compositionality metrics of (Chaabouni et al., 2020) are specifically made permutation-invariant wrt positions to reflect this. Moreover, whether adjacent tokens tend to be related is likely architecture-dependent, e.g. while this might hold for LSTM and GRU cells which have “recency” bias, it most likely not hold for Transformers which are permutation invariant (modulo positional embeddings).
* Eq (3) does not include conditional entropy. Do we need it monotonically decreasing (Q1) for HAS to hold?
* The paper shows that the word boundaries do not behave as one would expect and thus concludes that they are not “meaningful”. I wonder if it makes sense to check if, among the studied languages, there are individual words that encode specific attributes or values?
* Minor:
   * Typo: S2.2, first para: “HAS-baseThisboundary” ->  “HAS-base. This boundary”
   * I believe the text does not specify what kind of reward the agents get. Is it cross-entropy of the receiver’s output?



**Summary Of The Paper:**


This paper studies the languages that emerge between two agents that solve a task collaboratively. The main focus of the paper is on studying whether Harris’ articulation scheme (HAS) holds in the emergent languages. To test for that, the authors resort to an unsupervised segmentation algorithm derived from HAS. At its foundation, this algorithm compares the branching entropy before and at a position; it is assumed that at boundaries it starts growing.

Next, the problem of testing whether HAS holds is reduced to testing whether (Q1) Conditional entropy of the next symbol given the prefix is decreasing, (Q2) Branching entropy increases and decreases, (Q3) Derived boundaries are “meaningful”. (Q3) is further split into testing if the boundaries behave as expected when the number of attributes and values are changed.

The experimental study is performed using a standard setup where the sender agent receives a n_attr x n_val dimensional vector encoding a point in an attribute-value space with n_attr attributes each having n_val values. The sender then sends a multi-symbol message to the receiver agent which has to recover the original vector. Both agents receive a reward.

The experiments show mixed outcomes: (Q1) and (Q2) are indeed observed while the authors find the boundaries do not have the expected properties w.r.t. changing the number of attributes and values. Finally, the paper studies the relation between the Ease-of-Teaching and HAS and whether “words” obtained under segmentation follow ZLA.


**Summary Of The Review:**

Generally a clear paper with interesting experiments. My main comments can be summarized as:
* At this stage it is obvious that emergent languages are far from human languages from many points of view. Why is finding this particular difference useful?
* How studying word segmentation differentiates from studying compositionality of the emergent languages?
* Segmentation assumes that “words” are composed of adjacent tokens. Is it a natural assumption for the communication between artificial agents?

---

> ### Comment · Reviewer_BukJ · 2022-11-14
> **Comparing reviews**
>
> > We already have a strong understanding that human language-like,
> > interpretable features (ZLA, compositionality) often do not emerge by
> > themselves in emergent languages. For now, looking at HAS is motivated as yet
> > “another direction” to study emergent languages.
>
> It is certainly true that many works have investigated compositionality and ZLA
> in emergent language and found them to be absent in many cases (although
> I would argue it is not well _understood_).  This paper effectively supplements
> such prior work insofar as it moves beyond the paradigm of interpreting symbols
> in the emergent language as words (which itself is seldom explicitly
> justified).  It proposes instead that they might be a sub-word unit which could
> reveal previously unknown traits about compositionality, etc. in emergent
> language.  Although the ultimate result is negative, the technique, or
> derivatives thereof, can broaden how research looks at the structure of
> emergent language.
>
> > I think the paper could benefit from highlighting why it is particularly
> > worth studying.
>
> Although I think the paper definitely has sufficient motivation, I agree that
> its articulation could be expanded and improved.
>
> > Further, as formulated, the word boundaries segment the messages into groups
> > of adjacent tokens (ie. the 2nd token cannot be grouped with the 4th token
> > w/o being grouped with the 3rd token). Is there an intuition why this should
> > hold for the artificial agents? As far as I understand, any successful
> > language can be modified by a fixed permutation of positions w/o sacrificing
> > its expressiveness. For instance, the compositionality metrics of (Chaabouni
> > et al., 2020) are specifically made permutation-invariant wrt positions to
> > reflect this. Moreover, whether adjacent tokens tend to be related is likely
> > architecture-dependent, e.g. while this might hold for LSTM and GRU cells
> > which have “recency” bias, it most likely not hold for Transformers which are
> > permutation invariant (modulo positional embeddings).
>
> This is a great point that did not occur to me while reading the paper.  The
> paper does use an RNN architecture which means assuming contiguous segments is
> a reasonable first-order approximation.  I think this weakens the empirical
> results only slightly.  This could probably be addressed with some ablation
> studies which randomly shuffle (or deterministically permute) the emergent
> language and seeing if the _positive_ results still hold.  I think this would
> provide some insight into how significant the assumption of contiguous segments
> is.
>
> > How studying word segmentation differentiates from studying compositionality
> > of the emergent languages?
>
> Compositionality, generally speaking, concerns the nature of the mapping for
> words to meanings and whether the composition of words corresponds to the
> composition of meanings.  Most prior work assumes that emergent language
> symbols are the words and go from there.  This paper questions that assumption
> by investigating a different definition of "word" with respect to compositional
> semantics.

---

> ### Author Response · Authors · 2022-11-18
> **First Response to Reviewer zwe8**
>
> Thank you very much for reviewing our paper.
> We think that your questions are crucial and answering them would be beneficial for our paper.
>
> > Motivation: We already have a strong understanding that human language-like, interpretable features (ZLA, compositionality) often do not emerge by themselves in emergent languages. For now, looking at HAS is motivated as yet “another direction” to study emergent languages. I think the paper could benefit from highlighting why it is particularly worth studying.
>
> Regarding motivation, we agree with the point made by Reviewer BukJ. Previous studies implicitly rely on a one-message-per-word view or one-symbol-per-word view, without sufficient justification. Word segmentation must also be an important property of natural language but underexplored.
>
> In addition, we believe that one-message-per-word studies (e.g., ZLA) and one-symbol-per-word studies (e.g., compositionality, grammar) should be merged, at some point of the line of research that tries to bridge the gap between emergent and natural languages. One probable reason that hiders such a direction is that word segmentation is underexplored. We believe that our paper will be a catalyst for that direction.
>
> > In my view, the search for “word” boundaries seems to be closely related to the search of compositionality, i.e. whether there are elementary units within messages with a fixed “meaning” (Chaabouni et al., 2020). I believe the text could benefit from contrasting those pursuits. For instance, if compositionality in the sense of (Chaabouni et al., 2020) is achieved, would word boundaries become meaningful? or vice-versa?
>
> This is an important question.
> In our view, compositionality (in the sense of Chaabouni et al. 2020) and meaningful word segments (in the sense of HAS) are closely related but not strictly the same.
>
> We think that meaningful word segments have the following additional properties:
>
> 1. *Adjacency*: as you mentioned.
> 2. *Jittering redundancy*: the jittering behavior of branching entropy implies that the character redundancy repeatedly falls and rises. Suppose we obtain an emergent language that is perfectly compositional as well as optimal in length (i.e., efficient coding). Such a language does not have any segments, since the branching entropy does not jitter due to the optimality.
>
> On the other hand, meaningful segments do not necessarily mean perfect compositionality, either. In English, for example, “red,” “blue,” “green,” and “yellow” are meaningful segments that denote color attributes, while they differ in length. Length differences may hinder the perfect achievement of compositionality in the sense of Chaabouni et al. 2020.
>
> > Further, as formulated, the word boundaries segment the messages into groups of adjacent tokens (ie. the 2nd token cannot be grouped with the 4th token w/o being grouped with the 3rd token). Is there an intuition why this should hold for the artificial agents? As far as I understand, any successful language can be modified by a fixed permutation of positions w/o sacrificing its expressiveness. For instance, the compositionality metrics of (Chaabouni et al., 2020) are specifically made permutation-invariant wrt positions to reflect this. Moreover, whether adjacent tokens tend to be related is likely architecture-dependent, e.g. while this might hold for LSTM and GRU cells which have “recency” bias, it most likely not hold for Transformers which are permutation invariant (modulo positional embeddings).
>
> Assuming adjacency should be reasonable as long as we adopt RNN and auto-regressive decoder (for the sender agent).
> The dependence on the specific architecture is a limitation of our paper, and the revised version includes a discussion on that in the Discussion section (Section 7).
>
> We think that Chaabouni et al. 2020 is a beautiful study in the sense that it extracts the “disentanglement” issue from a complicated problem “compositionality” and it is independent of the “adjacency” issue.
> But, it seems the previous work can do so by implicitly relying on a one-symbol-per-word view.
>
> > Eq (3) does not include conditional entropy. Do we need it monotonically decreasing (Q1) for HAS to hold?
>
> Yes. The rising point of the branch entropy has special significance because it is, on average, decreasing.
>
> > I believe the text does not specify what kind of reward the agents get. Is it cross-entropy of the receiver’s output?
>
> Yes, it is the cross-entropy, following Chaabouni et al. 2020. This is unclear in the original version. In the revised version, we inserted “objective function” into  the beginning of “Architecture and Optimization,” Section 5.1:
> - We follow Chaabouni et al. 2020 for agent arthitectures, an objective function, and an optimization method.

---

### Official Review · Reviewer_BukJ · 2022-10-25

**Confidence:** 4
**Correctness:** 4
**Technical Novelty And Significance:** 4
**Empirical Novelty And Significance:** 2
**Recommendation:** 8

**Clarity, Quality, Novelty And Reproducibility:**

### Clarity
- The paper is well-organized and easy to follow.
  - The hypotheses are clearly stated such that they are falsifiable with a straightforward experiment.
    This is tough to find in emergent language papers.
- Although the pseudocode of Algorithm 1 is clear, the behavior is not entirely intuitive, so some sort of diagram or step-by-step example (even in the appendix) would be helpful.
- Section 2.1 mentions that HAS can be applied to non-phoneme sub-word units -- this is important to note early in the paper because I spent the first two pages thinking, "The components of emergent language messages do not really behave like phonemes nor are they intended to."
  - It would also be good to put the statement that "it is not even evident, in the first place, whether they have meaningful word segments" near the beginning.
    I agree with this statement, and I think its presence early on helps with the framing of the motivation, experiments, and results.
### Quality
- Overall, the hypotheses and experiments are very well suited to looking at whether or not HAS applies to (an) emergent language.
  - The experiments present an informative combination of positive and negative results which also provide a platform for future experimentation.
- It would be beneficial to have comparisons to HAS applied to natural language as showing that it works is more convincing and informative than just mentioning it.
  Ideally, we would have human data for an analogous task to the signalling game, but collecting such data is admittedly far beyond the scope of this paper.
  Instead, it might be helpful enough just to plot HAS performance on natural language in the same way that it is plotted for the emergent language data so as to have a point of comparison.
  - Along the same lines, ablation studies could also help give the real plots some perspective.
    This could even be as simple as, "what does it look like if I apply this analysis to a random language."
- There is no quantitative analysis for Q2 (rising and falling branching entropy) which limits the degree to which the experiments can support the hypothesis.
- The section on Zipf's law of abbreviation is mildly interesting but does not contribute to the overall paper.
  It seems more appropriate to put the section in the appendix and use the space for expanding upon previous discussion or just making the paper 0.3 pages shorter.
### Novelty
- The HAS algorithm makes minimal assumptions about the input language and relies on information theoretic notions both of which fit in with the constraints of analyzing emergent language.
- As the field of emergent language becomes more interested in complex structures of emergent language, understanding the nature of word segmentation in the messages of emergent language will become increasingly important.
- Much of the previous work on compositionality in emergent language treat each component of an emergent language message as its own word.
  Thus, this paper is novel in its approach of looking at these message components as sub-word units.
- Overall, this appears to be a natural and successful grounding of emergent language research in linguistic knowledge.
### Reproducibility
- The description of the experiments is clear.
- Since the code is not provided, I cannot comment on its quality or how easy it would be to reproduce directly.


**Strength And Weaknesses:**

### Strengths
- [major] Unsupervised word boundary detection is genuinely useful to studying the structures of emergent language and has not been tackled before.
- [major] The hypotheses and corresponding experiments are clear and effective.
- [major] This paper applies a concept from linguistics to emergent language in a fitting and relevant way.
- [minor] The empirical results of the experiments are informative vis-a-vis the hypotheses, although the results are not definitive or earth-shattering.
### Weaknesses
- [minor] No quantitative measure of branching entropy falling and rising
- [minor] Tough to put the empirical results in perspective due to lack of natural language/random language baseline or ablation study.
  Appendix A addresses this from one angle and helps make this a only a minor weakness.


**Summary Of The Paper:**

Zellig Harrs's articulation scheme (HAS) is an unsupervised method of discovering word segmentation based on the statistics of sub-word units in natural language.
This paper takes uses HAS on emergent language to determine whether the statistical properties of sub-word units in emergent language behave similarly to natural language.
The experiments show that while some of the low level statistical properties which HAS predicts are present, the evidence for HAS discovering meaningful word boundaries in emergent languages tested is largely negative.


**Summary Of The Review:**

I recommend accepting the paper because it applies a linguistic framework in natural language to emergent language which (1) is genuinely useful to studying the structures of emergent language and (2) is well-suited to the constraints which emergent language presents (e.g., few a prior assumptions of structure, comprising arbitrary symbols).
The experiments clearly test the hypotheses, and the empirical results are generally informative.
The places where the results do lack do not significantly detract from the aforementioned strengths.


### Misc.
- `2.2`: First sentence of 2.2 needs to be checked for typos.
- `2.2`: The little interlude about applying HAS-based boundary detection to natural language does not contribute much on its own.
  It can probably be removed if there is no further discussion of HAS applied to natural language.

---

> ### Comment · Reviewer_BukJ · 2022-11-14
> **Lowering empirical novelty and significance**
>
> In response to the other reviews and some comments from the area chair, I am lowering my "Empirical Novelty and Significance" score from 3 to 2, but I am keeping my Recommendation at an 8 for now because I believe the theoretical contributions to be the most important contribution.  The empirical results, while not definitively demonstrating some fact about emergent language, do illustrate at a basic level how HAS applies (or does not apply) to emergent language.

---

> ### Author Response · Authors · 2022-11-18
> **First Response to Reviewer BukJ**
>
> Thank you very much for giving a careful review, and even defending our paper!
> Your points are important for improving the clarity of our paper.
>
> > Although the pseudocode of Algorithm 1 is clear, the behavior is not entirely intuitive, so some sort of diagram or step-by-step example (even in the appendix) would be helpful.
>
> In the revised version, we added a simple illustrative figure for the boundary detection algorithm in Appendix F.
>
> > Section 2.1 mentions that HAS can be applied to non-phoneme sub-word units -- this is important to note early in the paper because I spent the first two pages thinking, "The components of emergent language messages do not really behave like phonemes nor are they intended to.”
>
> In the revised version, we inserted the following sentence into the 2nd paragraph, Section 1 (Introduction).
> - HAS holds not only for phonemes but also for other symbol units like characters.
>
> > It would also be good to put the statement that "it is not even evident, in the first place, whether they have meaningful word segments" near the beginning. I agree with this statement, and I think its presence early on helps with the framing of the motivation, experiments, and results.
>
> In the revised version, we inserted the statement into the beginning of the 3rd paragraph, Section 1 (Introduction).
>
> > it might be helpful enough just to plot HAS performance on natural language in the same way that it is plotted for the emergent language data so as to have a point of comparison.
>
> Fig 5 in [Tanaka-Ishii, 2005] and Fig 5 in [Tanaka-Ishii & Jin, 2008] might be helpful as an analogous equivalent for the qualitative example of Fig 4 in our paper.
>
> > There is no quantitative analysis for Q2 (rising and falling branching entropy) which limits the degree to which the experiments can support the hypothesis.
>
> We did not include quantitative analyses for Q2 in order to avoid duplication: the result of  Figure 5 (threshold=0, in particular) proves the jittering behavior of the branching entropy. However, some more detailed analyses may be interesting.
>
> > The section on Zipf's law of abbreviation is mildly interesting but does not contribute to the overall paper. It seems more appropriate to put the section in the appendix and use the space for expanding upon previous discussion or just making the paper 0.3 pages shorter.
>
> We agree with it, as we have to add a related work section and augment the discussion section, in response to the reviewers’ comments. In the revised version, All texts related to ZLA were moved to Appendix E.
>
> > The little interlude about applying HAS-based boundary detection to natural language does not contribute much on its own. It can probably be removed if there is no further discussion of HAS applied to natural language.
>
> We generally agree with you. But, at the same time, some readers might wonder if HAS really works for natural languages, in the first place. In the revised version, we moved the F-score reports in Section 2.2 into footnote 7.

---

### Comment · Reviewer_BukJ · 2022-11-14
**Responding to area chair's comments**

Responding to some points brought up to by the area chair.

> In particular, paper 816 mostly seems a negative result to me. While negative
> results are fine in principle, the bar is higher for them. It's relatively
> easy to get a negative result and there is always the question: did you try
> hard enough (and were you imaginative enough) to get a positive result or is
> the negative result really the final verdict on the question investigated.

I agree that the results are negative and not exceptionally strong, but I do
believe they are sufficient to support the theoretical contributions (i.e.,
connecting linguistically-grounded segmentation methods to natural language).

> A methodological problem with the paper is pointed out by reviewer xts9. The
> authors change elements of the emergent language paradigm, most importantly,
> words are no longer single characters. It is not clear to what extent this
> changes the standard emergent language paradigm. In other words, several
> variables are changed at the same time, which is not good methodology.

I am having a little trouble understanding this critique.  The authors are not
changing anything about the emergent language game itself but rather the
analysis thereof.  There is a bit of paradigm shift here going from seeing
a one-to-one correspondence betweens symbols and words, but I think this is
a strength of the paper since the one-to-one relationship is often just assumed
and not established.  In light of this, I am not sure what the multiple
variables being changed are.

> I personally am concerned about the fact that segmentation of written
> language is very limited cross-linguistically. In a lot of languages, you
> don't get natural segments through string segmentation because "underlying"
> segments change contextually. For example, in Spanish you have
>
>> yo hice / tu hiciste / ella hizo
>
> If you just segment, it's difficult to get the equivalence of hic and hiz.

If I understand correctly here, the stated difficulty is that observing the
orthography of language is not an effective means by which to derive
morphologically meaningful segments.  This is, in part, why the authors make
analogy to the segmentation of phonemes since those have a more direct
correlation with the identity of words than do graphemes, but maybe the
difficulty persists since phonological rules can cause variability in
phones/surface representations vs. the underlying representation.  Since the
symbols of the emergent language are the language itself (and not just an
approximation), it does not suffer from the same issues as writing down
a spoken language.  I may be missing the point of this critique, though.

> [Of course, you can get it with enough data, but that's the core problem of
> language that you never have enough data.]

Just an aside, but with emergent language, it is relatively simple to generate
arbitrary amounts of data since everything is programmatic (although emergent
languages, as they are now, are usually so simple as to not require large
amounts).

---

### Author Response · Authors · 2022-11-18
**General Response to Reviewers: About Paper Revision**

Thank you very much for carefully reviewing our paper!

We revised the paper as follows:
- moved all texts related to ZLA to Appendix,
- made the Related Work section (Section 8),
- added some limitations in the Discussion section (Section 7),
- fixed some typos, and
- made some minor changes (see individual replies).

---

### Decision · Program_Chairs · 2023-01-20

**Decision:**

Accept: poster

**Justification For Why Not Higher Score:**

See the list of weaknesses in the metareview

**Justification For Why Not Lower Score:**

The paper definitely makes a contribution that is worth publishing.

**Metareview: Summary, Strengths And Weaknesses:**

SUMMARY

The authors adopt the signaling game framework of
deep-learning based emergent communication models, in
particular, Chaabouni et al. (2020).  They investigate
whether Harris's articulation scheme (HAS) holds for
emergent languages in this framework.  HAS states that word
boundaries can be obtained solely from phonemes, i.e., that
the units that result from HAS are meaningful.  The
experiments suggest that they are not.  The authors conclude
that the currently used signaling game framework lacks
some necessary ingredients.

STRENGTHS

A major shortcoming of the signaling game framework of
emergent communication models is exposed.

Unsupervised word boundary detection helps understand the
structure of emergent languages and has not been extensively
studied before in this context (only one prior paper on the topic).

The paper can be argued to be partially based on linguistics
-- which everybody agrees is a good thing.

Well-executed study, clear writeup.

WEAKNESSES

The fact that a major shortcoming of the signaling game
framework of emergent communication models is exposed is a
strength, but it is a negative result. The bar for accepting
a paper whose main result is negative should be higher than
for positive results.

The paper does not provide the reason for why the emergent
communication model does not make HAS emerge.

The paper can be argued to be partially based on
linguistics, but the linguistics they actually use is a
questionable concept whose relevance for the study and
understanding of natural language is in doubt.

The paper is tied to one particular framework for emergent
languages and it is not clear to what extent the conclusions
hold for emergent languages in general.

The paper uses an RNN architecture. This means that we
assume contiguous segments. That is a reasonable first-order
approximation. However, it does not hold for emergent
languages in general (see reviewer zwe8, weaknesses).  This
means that the major conclusion (HAS does not hold for
emergent languages) comes with a caveat.

It is worrying that one of the two reviewers who clearly
advocate for the paper (recommendation 8) gives a low
empirical novelty/significance score of 2.



**Note From Pc:**

if the above contains the word "oral" or "spotlight" please see: "oral" presentation means -> notable-top-5% and "spotlight" means -> notable-top-25%. As stated in our emails, we are disassociating presentation type from AC recommendations